# Myofibroblast transcriptome indicates *SFRP2*<sup>hi</sup> fibroblast progenitors in systemic sclerosis skin

Tracy Tabib[1], Mengqi Huang [1], Nina Morse[1], Anna Papazoglou[1], Rithika Behera[1], Minxue Jia[2,3], Melissa Bulik[1], Daisy E. Monier[1], Panayiotis V. Benos [2,3], Wei Chen [4], Robyn Domsic[1] & Robert Lafyatis [1✉]

Skin and lung fibrosis in systemic sclerosis (SSc) is driven by myofibroblasts, alpha-smooth muscle actin expressing cells. The number of myofibroblasts in SSc skin correlates with the modified Rodnan skin score, the most widely used clinical measure of skin disease severity. Murine fibrosis models indicate that myofibroblasts can arise from a variety of different cell types, but their origin in SSc skin has remained uncertain. Utilizing single cell RNA-sequencing, we define different dermal fibroblast populations and transcriptome changes, comparing SSc to healthy dermal fibroblasts. Here, we show that SSc dermal myofibroblasts arise in two steps from an *SFRP2*<sup>hi</sup>/*DPP4*-expressing progenitor fibroblast population. In the first step, SSc fibroblasts show globally upregulated expression of transcriptome markers, such as *PRSS23* and *THBS1*. A subset of these cells shows markers indicating that they are proliferating. Only a fraction of *SFRP2*<sup>hi</sup> SSc fibroblasts differentiate into myofibroblasts, as shown by expression of additional markers, *SFRP4* and *FNDC1*. Bioinformatics analysis of the SSc fibroblast transcriptomes implicated upstream transcription factors, including *FOSL2*, *RUNX1*, *STAT1*, *FOXP1*, *IRF7* and *CREB3L1*, as well as *SMAD3*, driving SSc myofibroblast differentiation.

[1] Division of Rheumatology and Clinical Immunology, School of Medicine, University of Pittsburgh, Department of Medicine, Pittsburgh, PA, USA.
[2] Department of Computational and Systems Biology, School of Medicine, University of Pittsburgh, Pittsburgh, PA, USA. [3] Joint CMU-Pitt PhD Program in Computational Biology, Pittsburgh, PA, USA. [4] Division of Pulmonary Medicine, Allergy and Immunology, Department of Pediatrics, School of Medicine, University of Pittsburgh, Pittsburgh, PA, USA. ✉email: lafyatis@pitt.edu

Skin fibrosis is a prominent clinical feature in most patients with systemic sclerosis (SSc, otherwise known as scleroderma), and the defining clinical feature for stratifying patients into two major disease subsets, limited or diffuse cutaneous disease. Skin tightness and thickening lead to considerable morbidity related mainly to contractures of hands, as well as larger joints. It is also associated with pain, itching, and cosmetic anguish[1]. Clinically skin involvement in SSc is associated with thickening, tethering, tightness, and inflammation. Pathologically skin thickening is due to increased matrix deposition, most prominently type I collagen. Skin tightness may be due to this increase in matrix, but also correlates with the presence of myofibroblasts in the skin[2]. Thus, increased collagen production and the appearance of dermal myofibroblasts, typically seen first in the deep dermis, are pathogenic processes closely associated with the severity of clinical disease in SSc skin[3].

In many fibrotic diseases myofibroblasts are the main collagen-producing cell driving fibrosis (reviewed in ref. [4]). Perhaps more importantly in SSc skin, they exert tension on the tissue, and through this mechanism may contribute to skin and joint contractures[2,3,5]. TGF-β and cell tension are factors most strongly implicated in myofibroblast development[6,7]. Increasing matrix stiffness induces myofibroblast differentiation[8,9]. TGF-β also induces myofibroblast differentiation and α-smooth muscle actin (SMA), the product of the *ACTA2* gene and a robust though not specific marker of myofibroblasts in many different fibrotic diseases[10,11]. Matrix stiffness and myofibroblast contraction also activate TGF-β[12,13], setting up a reinforcing amplification signal for tissue fibrosis. Several cytokines mediate, synergize with, or are permissive for the effect of TGF-β on myofibroblast formation: CTGF/CCN2 (refs. [14,15]), endothelin-1 (ref. [16]), and PDGF[17–19]. Others, such as FGF2, inhibit myofibroblast formation[20,21], while yet others, such as IFNγ, activate or inhibit myofibroblast formation in different fibrotic models[22,23].

Defining the phenotype of myofibroblasts beyond their expression of SMA has been challenging due to a rudimentary understanding of fibroblast heterogeneity in general and a paucity of specific markers of different fibroblast populations. However, recent studies have shed light on fibroblast heterogeneity in both mice and humans. In mice, markers are stable or dynamic (typically downregulated in adult mice) for dermal papilla (*CRABP1*), papillary (*DPP4/CD26*), and reticular (*PDPN*, SCA1/ATXN1) fibroblasts[24,25]. Other investigators found that engrailed/*DPP4*-expressing fibroblasts in murine skin are profibrotic[26]. We have recently described two major and five minor fibroblast populations in normal skin[27]. The most common dermal fibroblast is long and slender, and expresses *SFRP2* and *DPP4*. This population can be further divided into fibroblast subpopulations selectively expressing *WIF1* and *NKD2*, or *CD55* and *PCOLCE2*. A second major fibroblast population expresses *FMO1* and *MYOC*. Minor fibroblast populations express *CRABP1*, *COL11A1*, *FMO2*, *PRG4*, or *C2orf40*. *CRABP1*-expressing fibroblasts most likely represent dermal papilla cells and *COL11A1*-expressing cells most likely represent dermal sheath cells, but *FMO2*, *PRG4*, and *C2orf40* minor fibroblast subsets are uncharacterized. Other recent studies have shown markers that distinguish between papillary (*COL6A5*, *APCDD1*, *HSPB3*, *WIF1*, and *CD39*) and reticular (*CD36*) dermal fibroblasts[28].

Myofibroblasts are currently best defined by SMA staining; however, SMA is expressed by a variety of other cell types, including smooth muscle cells (SMCs), myoepithelial cells, pericytes, and dermal sheath fibroblasts. These other cell types can be distinguished by expression of additional markers: desmin (*DES*) and smoothelin (*SMTN*) for SMCs[29]; regulator of G protein signaling 5 (*RGS5*), chondroitin sulfate proteoglycan 4 *CSPG4/NG2*, platelet-derived growth factor receptor beta (*PDGFRB*) for

pericytes[30]; keratin 5 (*KRT5*) and keratin 14 (*KRT14*) for myoepithelial cells. However, there are no uniformly accepted specific markers for myofibroblasts. Cadherin 11 (*CDH11*) expression is associated with myofibroblast development and implicated in contractile force across myofibroblasts[31], but *CDH11* is expressed more diffusely by fibroblasts, as well as by macrophages in SSc skin[32]. Thus, we lack specific markers for myofibroblasts and have limited understanding of the origins of this key pathogenic cell type in SSc skin.

The cellular progenitors of myofibroblasts in fibrotic disease models have been the source of increasingly sophisticated lineage tracing studies in mice, revealing that a variety of cell types can convert into myofibroblasts, including pericytes, epidermal cells, endothelial cells, preadipocytes, as well as fibroblasts (reviewed in[4]). In murine renal fibrosis, it appears that myofibroblasts arise from multiple progenitor cell types, including resident fibroblasts and bone marrow-derived cells, or transition from endothelial and epithelial cells[33]. Other lineage tracing studies have emphasized perivascular cells as progenitors of myofibroblasts in skin and muscle wound scarring[34]. In bleomycin-induced skin fibrosis adiponectin-expressing cells in adipose tissue can act as myofibroblast progenitors. In this model of SSc, transient cells co-expressing perilipin and SMA precede the development of myofibroblasts[35]. Although these studies suggest that multiple cell types can serve as progenitors of myofibroblasts in murine models, their origin in human disease, including SSc, has remained obscure. A recent study has shown that resident CD34+ fibroblasts in SSc skin undergo a change in phenotype characterized by downregulated CD34 expression and upregulated podoplanin (*PDPN*) expression[36]. Although this phenotypic change was not strongly associated with the presence of myofibroblasts, markers of these cells were retained on myofibroblasts, suggesting that resident CD34+ dermal fibroblasts may be the precursors of myofibroblasts in SSc skin.

Here, we show the transcriptome–phenotypic changes of fibroblasts that occur in the skin from patients with SSc, focusing on identifying myofibroblasts using single-cell RNA-sequencing (scRNA-seq) on total skin cell digests from SSc and healthy controls subjects.

## Results

**Single-cell transcriptomes from control skin.** We have previously described fibroblast heterogeneity in normal skin[27]. We reanalyzed fibroblasts from normal healthy skin using an updated clustering algorithm and four additional discrete skin samples (Supplementary Figs. 1–5). We observed the same cell populations we described previously based on the top differentially expressed genes (Supplementary data 1), but some additional populations were also apparent. *SFRP2/DPP4*-expressing fibroblasts, long narrow cells representing the most common population of fibroblasts, as before divided into two groups of cells: a *WIF1/NKD2*-expressing subgroup (cluster 1), also expressing *HSPB3*, *APCDD1*, and *COL6A5*, previously identified as markers of papillary dermis[28], and a *PCOLCE2/CD55/SLPI*-expressing subgroup (cluster 0, Supplementary Figs. 1, 2, 3A, B and 4). A second major population, expressing *APOE*, included *MYOC/FMO1* fibroblasts described previously[27], expressing low levels of *APOE* (*APOE*lo/*MYOC*, cluster 3), as well as a two subpopulations expressing higher levels of *APOE*, one of which also expressed high levels of *C7* (*APOE*hi/*C7*, cluster 4), the other a *APOE*hi/*C7* subset that also expressed high levels of *CCL19*, appearing mainly around vascular structures (*APOE*hi/*C7/CCL19*, cluster 7; Supplementary Figs. 3A, B and 4). We identified several other cell populations based on previously described murine and human fibroblast markers surrounding hair follicles: *CRABP1/COCH*-expressing dermal papilla (cluster 5) and *COL11A1/ACTA2*-expressing dermal sheath cells

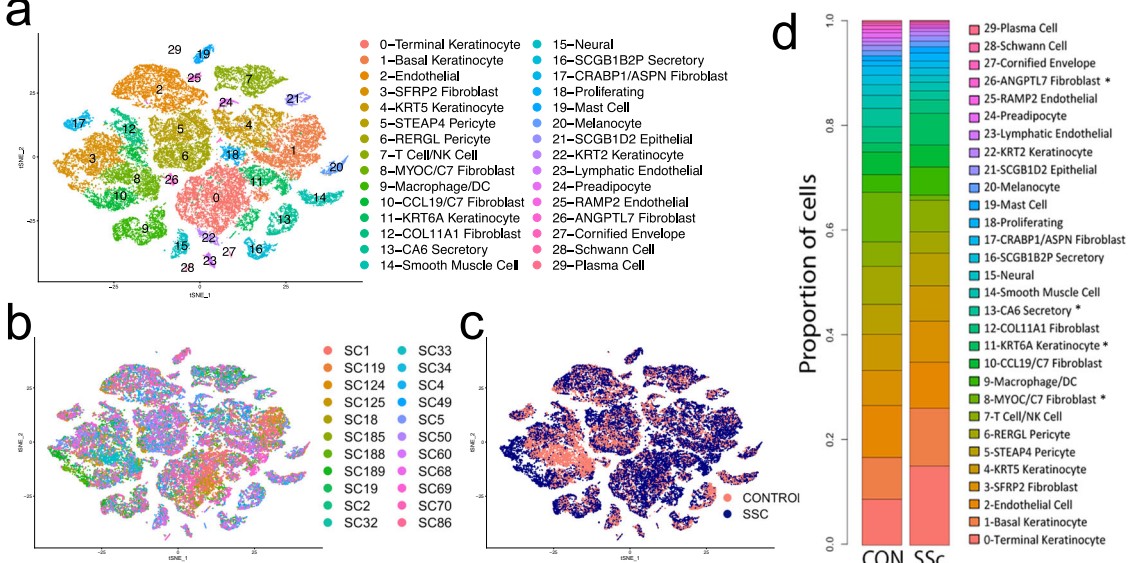

**Fig. 1 T-SNE plot of scRNA-seq data from control and SSc skin biopsies.** Transcriptomes of all cells obtained after enzymatic digestion of dorsal mid-forearm skin biopsies from 10 healthy control and 12 SSc subjects, showing each SLM cluster by color (**a**) or by source from each patient (**b**) or by source form SSC (blue) or control (red) biopsies (**c**). The proportion of cells in each cluster is indicated (**d**). Cell populations are differentially expressed between the SSc and control groups (p < 0.001, chi square test). Stars (*) indicate different proportions between SSc and control subjects, (p < 0.05).

(cluster 9, Supplementary Figs. 3A, B and 4)[25,37]. Two other cell populations cluster adjacent to dermal sheath and dermal papilla fibroblasts. One appears in the papillary dermis based on POSTN immunohistochemical staining (*ASPN/POSTN*, cluster 2, Supplementary Fig. 4). The other expressed high levels of *PTGDS* (cluster 6), though this was not a specific marker. Finally, two small discrete populations of fibroblasts expressed *SFRP4/ANGPTL7* (cluster 8) and *SFRP4/LINC01133* (subset of cluster 3). We have previously described *SFRP4*-expressing fibroblasts in normal papillary dermis[38]. In our recent scRNA-seq description of normal fibroblast populations, we also described *PRG4*+ fibroblasts[27]. Although these cells did not form a discrete cluster on this reanalysis *PRG4*-expressing cells could be seen to group within *PCOLCE2*+ fibroblasts (not shown).

**Single-cell transcriptomes from SSc and control skin.** We then compared scRNA-seq of single-cell suspensions from mid-forearm skin biopsies of 12 discrete samples from patients with SSc with the 10 control mid-forearm biopsy data described above. Control and SSc patients were balanced across sex (control = 5/10 female; SSc = 7/12 female), age (control mean age 51.9, median age = 57.5; SSc mean age = 54.7, median age = 57.5; Supplementary Data 2). Similar numbers of cells were obtained from control (mean = 2821.6 cells/biopsy and median = 2623 cells/biopsy) and SSc (mean 3082 cells/biopsy and median = 3267 cells/biopsy). All patients with SSc had diffuse cutaneous disease with a mean MRSS = 26.1, median MRSS = 25. Disease duration was variable, between 0.48 and 6.48 years. Several of the patients were taking disease-modifying medications, as indicated (Supplementary Data 2).

Cell transcriptomes were clustered by *t*-distributed stochastic neighbor embedding (*t*-SNE), revealing all expected skin cell types, identified by examining the top differentially expressed genes in each cluster (Fig. 1a and Supplementary Data 3). Cell types in clusters were similar to cell types seen in normal skin (Fig. 1a, and Supplementary Figs. 6 and 7, ref. [27]). Cells from each subject (Fig. 1b) and chemistry (Supplementary Fig. 7) were distributed in each cluster. The proportion of each cell population

was generally preserved between healthy and SSc skin; however, showing some changes in fibroblast and keratin 6A-expressing keratinocyte populations (Fig. 1d). Notably, even at this low resolution, SSc fibroblasts can be seen to cluster separately from control fibroblasts, whereas for other cell types SSc and control cells largely overlie each other (Fig. 1c). UMAP clustering of cells gave similar results (Supplementary Figs. 9–11). We compared the average change in gene expression by SSc to healthy cells in each cluster (Supplementary Data 4)

**Dermal fibroblast heterogeneity is preserved in SSc skin.** We selected the cell clusters of fibroblasts based on expression of *COL1A1*, *COL1A2*, and *PDGFRA* (clusters 3, 8, 10, 12, 17, and 26 from Fig. 1a), as we described previously these genes to be robust fibroblast cluster markers [27]. We reanalyzed just these cells by UMAP, revealing ten fibroblast cell types (Fig. 2a), generally paralleling those found in normal skin (see above and ref. [27]). Fibroblast subclusters included cells from each subject (Fig. 2b). However, fibroblasts from SSc patient skin samples showed prominent shifts between clusters (Fig. 2c). Each subcluster could be identified by characteristic gene expression of top differentially expressed genes (Figs. 2d and 3a, Supplementary Fig. 12, and Supplementary Data 5).

Fibroblasts expressing high levels of *SFRP2* (*SFRP2*hi fibroblasts, Figs. 2a and 3a, fibroblast subclusters 1, 3, and 4), represent the major population of dermal fibroblasts, which are long slender cells found between collagen bundles[27]. *SFRP2*hi fibroblasts included three subpopulations: subpopulations expressing *WIF1* and *NKD2* (*WIF*+ fibroblasts, subcluster 3) and *SLPI*, *PCOLCE2*, and *CD55* (*PCOLCE*+ fibroblasts, subcluster 1) found previously in normal skin (Figs. 2a and 3a), and a new subcluster of cells found mainly in SSc skin fibroblasts (*PRSS23*+ fibroblasts, subcluster 4; Figs. 2a and 4a). Top and highly statistically significant GO terms associated with this new cluster were extracellular matrix organization and extracellular structure organization (completely overlapping GO terms); collagen fibril organization; response to wounding; and skeletal system development (Supplementary Data 6).

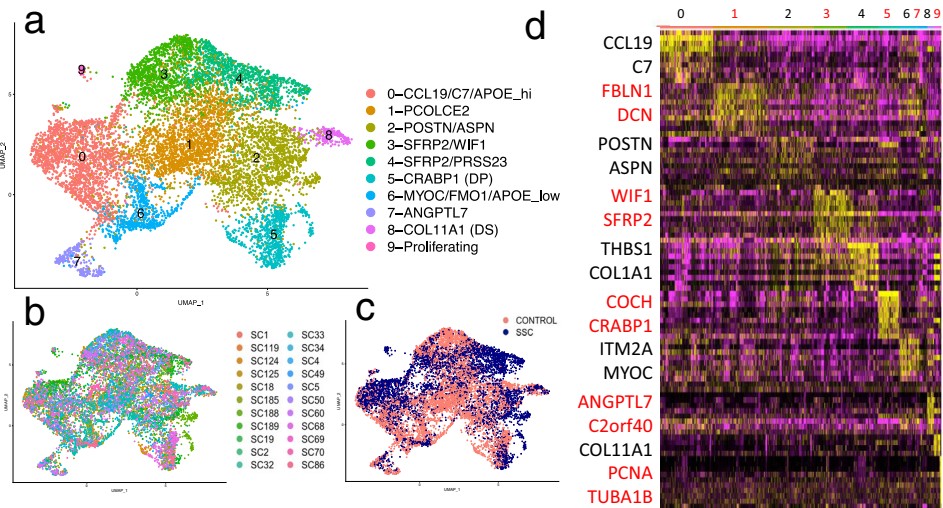

**Fig. 2 UMAP plot of scRNA-seq reclustering of fibroblasts and heatmap of subclusters.** UMAP analysis of transcriptomes of fibroblasts (clusters 3, 8, 10, 12, 17, and 26 from Fig. 1) from 10 healthy control and 12 SSc subjects, showing each SLM cluster by color (**a**) or by source from each patient (**b**) or by source form SSC (blue) or control (red) biopsies (**c**). Clustering of showing most differentially expressed genes associated with UMAP clusters. Yellow indicates increased expression, purple lower expression. Key marker genes are enlarged to the left (**d**).

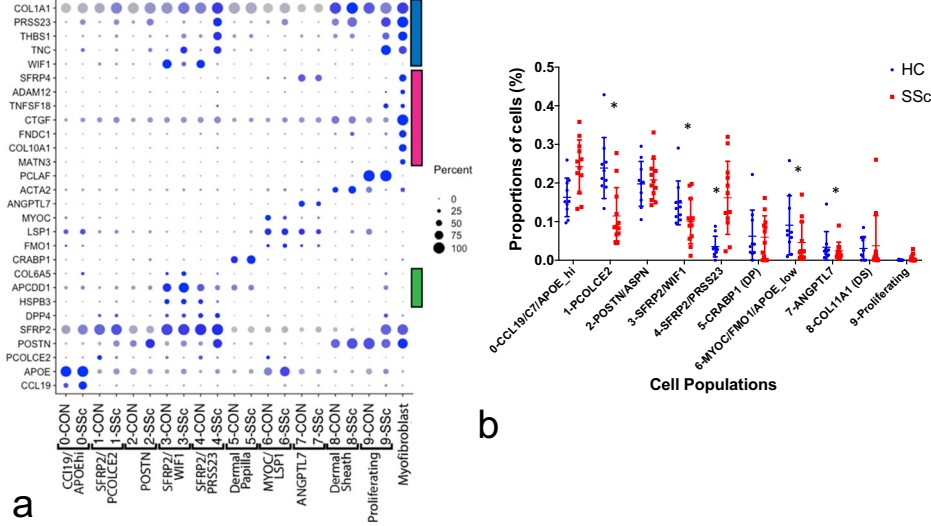

**Fig. 3 Key marker genes and proportions of fibroblast subclusters from control healthy (*n* = 10) and SSc (*n* = 12) skin.** Dot plots of gene expression markers of fibroblasts populations in healthy and SSc skin (**a**). Subpopulations of fibroblasts, including dermal sheath, dermal papilla, papillary (green bar), reticular and SSc fibroblasts (blue bar), and myofibroblasts (red bar), are indicated. Proportions of fibroblast subclusters as from healthy control (HC, blue) numbered and SSc biopsies (red; clusters are numbered as in Fig. 2) (**b**). Cell populations are differentially expressed between the groups ($p < 0.001$, chi square test), and clusters 1, 3, 4, 6, and 7 showed different proportions of cells comparing SSc and HC subjects ($p < 0.05$, bars = mean and error bars = standard deviation).

Expression of *APOE* defined cells in two clusters: *APOE*hi/*CCL19/C7*-expressing fibroblasts (clusters 0), and *APOE*low/*FMO1/MYOC*-expressing cells (subcluster 6, Figs. 2a and 3a). We have previously identified this latter population fibroblast population as distributed in interstitial and perivascular regions[27]. The larger subpopulation of cells (subcluster 0) included a subgrouping of cells, highly expressing *CCL19* showing a strong trend toward more SSc fibroblasts (Fig. 3b), the SSc *CCL19*+ fibroblasts clustering separately from the control *CCL19*+ fibroblasts (Fig. 2c), expressing higher levels of *CCL19* (Fig. 3a) and localizing primarily perivascularly (Supplementary Fig. 4).

Three adjacent clusters showed markers of cells associated with hair follicles (subclusters 2, 5, and 8). *CRABP1*-expressing cells

likely represent dermal papilla fibroblasts (DP, subcluster 5) and *ACTA2/SOX2*-expressing cells likely represent dermal sheath fibroblasts that may include dermal sheath stem cells (DS, cluster 8, refs. [37,39]). Cells in cluster 2 appear to represent cells closely related to hair follicles and/or papillary fibroblasts, as *ASPN* and *F2R* expressed by cells in this cluster stain brightly cells surrounding hair follicles (ref. [40] and see Human Atlas online), and *POSTN* stains brightly in the papillary matrix (Supplementary Fig. 4). The close relationship between these cells is consistent with the observation that papillary dermal fibroblasts are required to regenerate hair follicles[24]. Other markers of papillary dermal cells[28]: *APCDD1, HSPB3,* and *COL6A5* were expressed by cells in subcluster 3 (marked by a green bar in

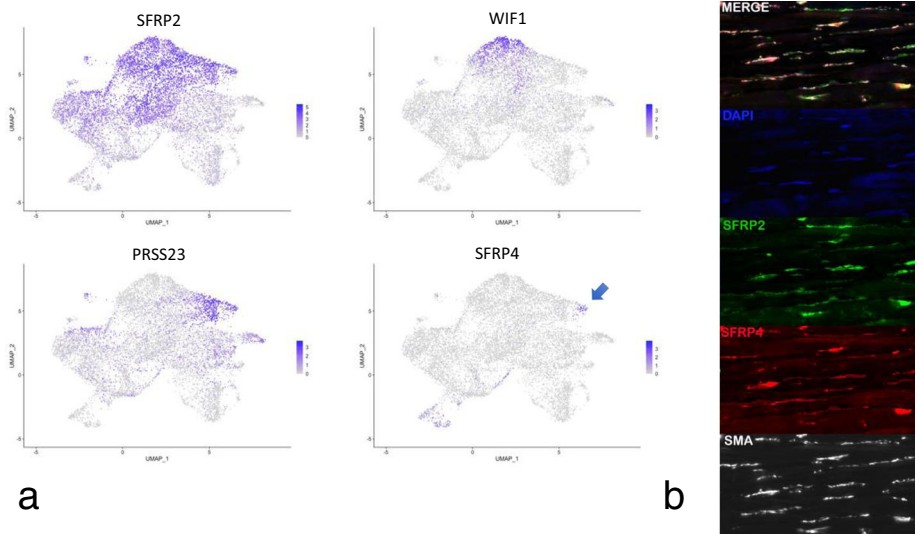

**Fig. 4 Gene expression and immunofluorescent staining of fibroblasts and myofibroblasts in SSc skin.** Cells from combined analysis of healthy control (n = 10) and SSc (n = 12) mid-forearm skin biopsies. Purple indicates increased expression (**a**). SSc SFRP2+ fibroblasts express increased PRSS23, decreased WIF1, and the myofibroblast subpopulation expresses SFRP4. Deep dermis from a patient with diffuse cutaneous SSc co-stained with SFRP2 (green) and SFRP4 (red), SMA (white) show strong overlap between staining of myofibroblasts with SFRP2 and SFRP4 (**b**). Nuclei (purple) are stained with Hoechst. Staining representative of n = 5. Scale bar = 50 μM.

Fig. 3a), part of the *SFRP2*-expressing population found also in the reticular dermis.

Our previous studies show *SFRP4*-staining fibroblasts in the papillary dermis of healthy, as well as SSc skin[38]. Thus, *ANGPTL7/C2orf40/SFRP4*-expressing cells (subcluster 7, Figs. 2a and 3a) represent a population of papillary fibroblasts, a second *SFRP4*+ population found only in SSc skin, characterized below, representing myofibroblasts (Fig. 3a).

Collectively these studies indicate that the papillary dermis includes several different fibroblast populations, as they are found in subclusters 2, 3, and 7.

**SSc fibroblasts show global alterations in phenotype.** Strikingly most of the fibroblasts from SSc patients clustered separately from the control subjects on UMAP dimensional reduction (Fig. 2c). SSc fibroblasts clustered prominently in cluster 4 (*SFRP2*hi/*PRSS23*+ fibroblasts) and also in a discrete region within cluster 0 (*CCL19*+ fibroblasts, Supplementary Fig. 12). These two clusters showed proportionately more cells originating from SSc compared to healthy, control biopsies (Fig. 3b). Reciprocal changes in cell proportions were seen in clusters 1, 3, and 6 with greater proportions of control cells in these clusters. In contrast to the marked separation between SSc and normal fibroblasts in the clusters above, fibroblasts predicted to reside in the papillary dermis, and DP and DS fibroblasts associated with hair follicles (subclusters 2, 5, and 8) were distributed in an approximately equal proportion between SSc and normal samples (Figs. 2c and 3b). Together these results indicated a widespread shift in the phenotype of at least two different fibroblast populations in SSc reticular dermis, but not in fibroblast populations associated with the hair follicle and papillary dermis.

*SFRP2*hi/*WIF1*+ fibroblasts (subcluster 3) were largely depleted in SSc skin with the appearance of *SFRP2*hi /*PRSS23*+ fibroblasts in the adjacent (subcluster 4). Comparing these two clusters directly revealed top differentially expressed genes, including *COMP* and *THBS1*, genes highly associated with the MRSS and previously identified as biomarkers of skin disease[41,42] (Supplementary data 7).

**SSc fibroblasts show discrete altered gene expression.** To investigate the changes in the transcriptome–phenotype of fibroblasts in SSc skin, we compared gene expression between SSc and control fibroblasts in each cluster (Supplementary data 8). Several of the highly upregulated genes in cluster 4 were recognizable as genes previously shown to correlate with the severity of SSc disease, such as *THBS1* (refs. [41,42]), *TNC*[43], *CTGF*[42], *THY1* (ref. [36]), *CDH11* (ref. [32]), and *CCL2* (ref. [44], Supplementary Data 8). However, particularly striking to us was the marked upregulation of *SFRP4*, a gene we had studied previously in the context of a putative role for Wnts in SSc[38]. Further, on examining these and other genes increased in SSc *SFRP2*hi/*PRSS23*+ (cluster 4) fibroblasts, we broadly observed two patterns of expression. Either genes were expressed by most cells in this cluster, such as *PRSS23*, *THBS1*, and *TNC*, or they were expressed by a subset of cells in this cluster, such as *SFRP4*, *ADAM12*, *TNFSF18*, *CTGF*, *FNDC1*, *COL10A1*, and MATN3 (Figs. 3a and 4a). We did not see any suggestion of preadipocyte, pericyte, or myeloid markers in myofibroblasts to suggest a transition from these cell types (Supplementary Fig. 13).

**Myofibroblasts co-express *SFRP2* and *SFRP4*.** We showed previously that *SFRP4* is upregulated in SSc skin and stains cells in the deep dermis, and that staining correlates with the MRSS[38]. However, at the time we did not associate this staining with myofibroblasts. Based on our scRNA-seq data showing a discrete cluster of *SFRP2*hi/*SFRP4*+ fibroblasts, we co-stained SFRP4 with SMA, the best-defined marker of myofibroblasts. We found that these two markers co-stain myofibroblasts in the deep dermis (Fig. 4b and Supplementary Fig. 14). We have recently shown that SFRP2 stains long, thin cells in normal dermis[27]. Here, we show that SMA staining myofibroblasts co-stain with SFRP2, and that *SFRP4*-expressing cells also co-stain with SFRP2, indicating that *SFRP2/SFRP4* co-expressing cells represent SSc dermal myofibroblasts.

**Myofibroblasts show a discrete transcriptome.** We compared gene expression of *SFRP2*hi/*PRSS23*+/*SFRP4*− fibroblasts to

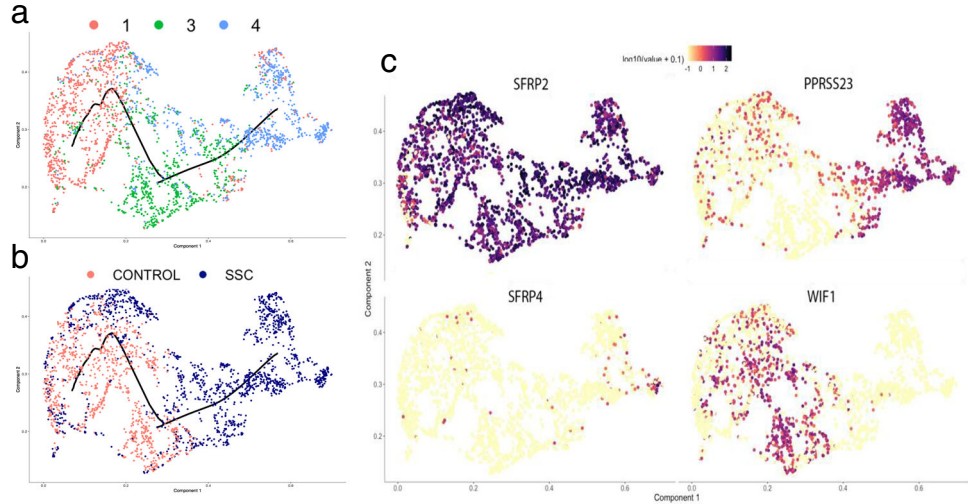

**Fig. 5 Pseudotime modeling of *SFRP2*+ fibroblast differentiation in SSc skin.** Fibroblast subclusters defined in Fig. 2 were analyzed using Monocle with the trajectory as indicated by the black line, with cells colored by subcluster of origin: subclsuter 1 (red), subcluster 3 (green), and subcluster 4 (blue; **a**) or by subject status: healthy control (red) and SSc (blue; **b**). *SFRP2* was expressed by all of the cells (**c**, right lower panel). *PRSS23* was expressed more highly by cells clustered later in pseudotime, corresponding to fibroblast subcluster 4 (**c**). *SFRP4* was expressed more highly expressed even later in pseudotime, corresponding to myofibroblasts identified in *t*-SNE plots (Figs. 3a and 4a).

*SFRP2*[hi]/*PRSS23*+/*SFRP4*+ myofibroblasts. The *SFRP2*[hi]/*SFRP4*+ fibroblasts were composed mostly of SSc cells (84/85 cells). Genes in addition to *SFRP4* that are regulated (Supplementary data 9) included several other genes associated with the WNT pathway: *SFRP1* and *WNT2*, and *ACTA2*, the gene encoding SMA (expressed 4.55-fold more highly in *SFRP2*[hi]*SFRP4*+ fibroblasts, Supplementary Data 9).

**SFRP2[hi]WIF1+ fibroblasts are progenitors of myofibroblasts.** To further investigate the relationship between *SFRP2*[hi] fibroblasts from healthy control skin, and fibroblasts and myofibroblasts in SSc skin, we used Monocle, an algorithm that tracks the relationship between single-cell transcriptomes known as pseudotime [45]. Pseudotime analysis indicated that there is a linear progression from *SFRP2*[hi]*PCLOCE2*+ fibroblasts (subcluster 1) to *SFRP2*[hi]*WIF1*+ fibroblasts (subcluster 3) to *SFRP2*[hi] *PRSS23*+*WIF1*− fibroblasts (subcluster 4) to *SFRP2*[hi]*PRSS23*+ *SFRP4*+ myofibroblasts (Fig. 5a, b). Although there is no polarity to the pseudotime analysis, since myofibroblasts are not present in normal skin, they most likely represent a later time in differentiation. This analysis indicates that *SFRP2*[hi]*PRSS23*+*WIF1*− fibroblasts are the immediate progenitors of myofibroblasts, and *SFRP2*[hi]*WIF1*+ fibroblasts the progenitors of *SFRP2*[hi]*PRSS23*+*WIF1*− fibroblasts.

This analysis reinforced the upregulated gene expression, transcriptome markers identified by examining the transcriptomes of *SFRP2*[hi]*SFRP4*+ myofibroblasts. These included *COL10A1*, *FNDC1*, *SERPINE1*, *MATN3*, and *CTGF* (Supplementary Fig. 15).

To further support the trajectory of *SFRP2*-expressing fibroblasts, we applied Velocyto, analyzing the scRNA-seq data based on spliced and unspliced transcript reads [46], supporting movement of *SFRP2*[hi]*WIF*+ to *SFRP2*[hi]*PRSS23*+*WIF1*− fibroblasts to *SFRP2*[hi]*PRSS23*+*SFRP4*+fibroblasts (Supplementary Fig. 16).

**Increased proliferating SFRP2[hi]PRSS23+WIF1− fibroblasts in SSc skin.** A minor population of fibroblasts (subcluster 9), clustered separately from the other fibroblasts because they highly differentially expressed genes associated with cell proliferation

(including *PCNA* and *PCLAF*, Fig. 6a). We have shown previously that macrophages in IPF lungs expressing these markers are indeed proliferating cells [47], though in this case these are extremely rare (representing only 0.38% of the fibroblasts and 0.080% of the total cells), and thus unlikely to be detected by immunohistochemistry. Of the 39 cells, only 2 of the cells in this subcluster were from healthy skin. The other 37 cells were from the SSc skin samples. The two cells from healthy skin showed markers of dermal sheath cells (*DPEP1* and *COL11A1*, Fig. 6b). In contrast, all of the cells from the SSc patients expressed markers of *SFRP2*[hi]*PRSS23*+*WIF1*− cells (*SFRP2*, *PRSS23*, *TNC*, and *COL10A1*). However, these cells did not selectively express markers associated with differentiation of *SFRP2*[hi]*PRSS23*+*WIF1*− cells into myofibroblasts (not shown).

**Correlation between bulk microarray and SFRP2[hi]/SFRP4+ cells.** Several previous studies have examined bulk mRNA expression in SSc skin [42,48–50]. Analyzing microarray data from our previous biomarker study and clinical trials conducted by our center using the same microarray platform [42,51–53], we compared genes upregulated in SSc whole biopsy gene expression data with our single-cell results. Several microarray clusters showing genes upregulated in subsets of SSc patients contained genes expressed more highly by SSc fibroblasts or myofibroblasts, emphasizing the important role these genes play in these signatures (Fig. 7a). Probing the single-cell dataset with these clusters as gene modules showed that, indeed, they detect the global change in SSc fibroblasts (PRSS23 signature), or the change associated with myofibroblasts (SFRP4 signature) or (COL10A1 signature, Fig. 7b). We have shown in previous publications that expression of several of the genes in these clusters (*THBS1*, *COMP*, *ADAM12*, and *CTGF*) correlate highly and statistically significantly with the MRSS [42], so the observation that these genes cluster together in bulk RNA-seq analysis and their co-expression in our scRNA-seq dataset in the transition of healthy *SFPR2*[hi] fibroblasts to SSc *SFRP2*[hi] fibroblasts (*THBS1*) and myofibroblasts (*ADAM12* and *CTGF*) is consistent with the roles of these SSc fibroblast populations in driving clinical disease. Expression of *PRSS23*, a marker for the first step in SSc fibroblast differentiation, correlated highly with the MRSS (Fig. 7c), confirming that the first step in SSc fibroblast differentiation is associated with clinical skin disease.

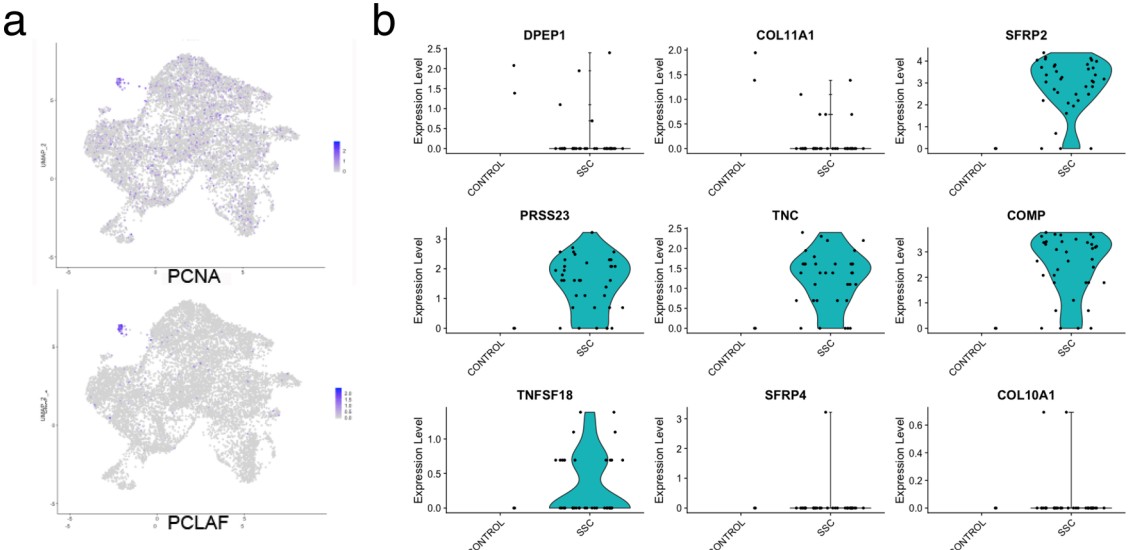

**Fig. 6 Proliferating fibroblasts in healthy control and SSc skin.** Feature plots indicating that expression of proliferation markers, *PCNA* and *PCLAF*, are limited to cells in cluster 9 (**a**, **b**). Violin plots indicate gene expression by proliferating cells (subcluster 9), showing markers of dermal sheath cells (*DPEP1* and *COL11A1*) by healthy control cells (2 cells) and markers of *SFRP2*hi*PRSS23+WIF1−* cells (*SFRP2*, *PRSS23*, *TNC*, *COMP*, and *TNFSF18*) by SSc fibroblasts (37 cells). Only one SSc fibroblast expressed markers of myofibroblasts (*SFRP4* and *COL10A1*).

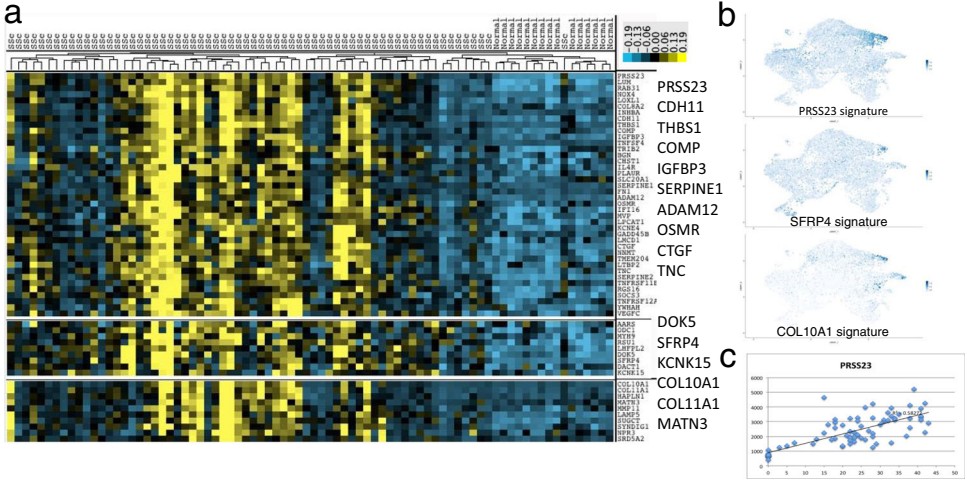

**Fig. 7 Bulk RNA expression data clusters reflect gene expression by SSc fibroblasts and myofibroblasts.** Bulk gene expression from microarrays of patients with SSc ($n = 66$) and healthy control skin ($n = 9$) was clustered hierarchically (**a**; yellow = high, blue = low expression). Feature plots corresponding to gene expression signature in of each microarray cluster are shown (**b**, Seurat AddModuleScore function). The correlation of PRSS23 with the MRSS is shown (**c**).

**Predicted transcription factor regulation of SSc fibroblast and myofibroblast gene expression**. A significant challenge to gaining further insight into disease pathogenesis is relating gene expression changes to underlying alterations in intracellular signaling. To address this question, we analyzed our data using SCENIC, a computational method developed for detecting transcription factors (TFs) networks [54]. *T*-SNE analysis of subclusters 1–4 by regulon (rather than by gene) showed a clear separation of the SSc fibroblasts from subcluster 4 (Fig. 8a, b). Clustering the TFs driving regulons plots revealed a series of TFs, including *TGIF2*, *FOSL2*, *RUNX1*, *STAT1*, and *IRF7* (Fig. 8c, d), these genes expressed more highly also in this cell population (Fig. 8e).

To examine TFs associated with myofibroblast differentiation more selectively, we divided the *SFRP2*hi*WIF1−SFRP4+* (myofibroblast populations) from the *SFRP2*hi*PRSS23+* (remainder of subcluster 4, composed mainly of SSc fibroblasts) and compared

the TFs regulating these two subclusters with scRNA-seq subcluster 3, *SFRP2*hi*WIF1+* fibroblasts, composed of both healthy and SSc cells (Supplementary Fig. 17). Clustering of the TFs from this analysis showed several of the same TFs predicted as driving subcluster 4 differentiation (*FOSL2*, *FOXP1*, *RUNX2*, *RUNX1*, and *IRF7*), as well as several other TFs (Supplementary Fig. 18). We observed similar TFs if first filtering the inputted genes requiring six UMI and expression in 1% of cells (Supplementary Fig. 19). In addition, to correct for potential overfitting due to the variable number of cells per cluster, we downsampled subclusters 3 and 4 40 times and iteratively analyzed predicted TFs by SCENIC. Regulons of *IRF7*, *STAT1*, and *CREB3L1*, upregulated in 38 of 40 downsamplings, further validated these TFs in myofibroblast differentiation (Fig. 8e).

Surprisingly, none of the transcriptomes in the initial analysis, including all transcriptome genes associated with each subcluster

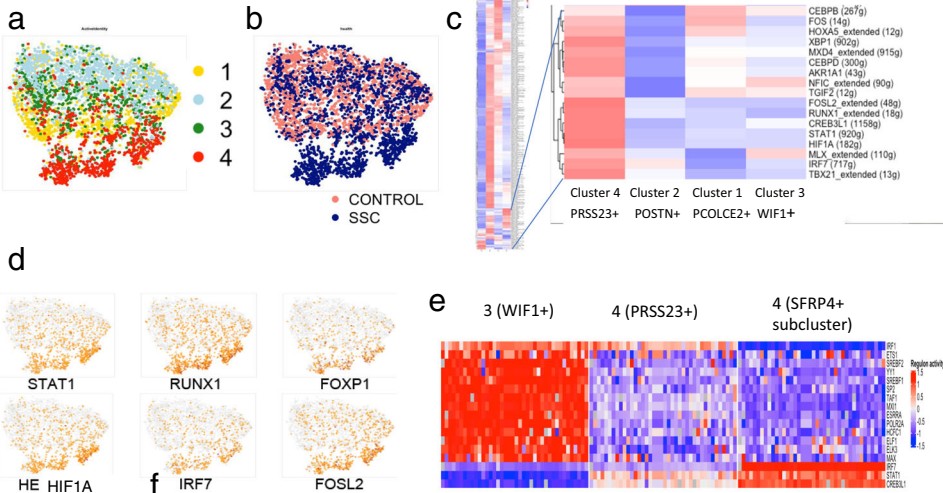

**Fig. 8 Regulons associated with TFs.** Clustering of fibroblast subclusters 1–4 by regulon expression, colored according to gene expression subclusters as in Fig. 2 and indicated in the legend (**a**) and according to SSc/healthy disease status (**b**). Clustering of regulons identified comparing fibroblast subclusters 1–4, expanded section showing regulons upregulated in cluster 4 (**c**). Gene expression indicated on regulon *T*-SNE plots of select TFs (**d**, brown = increased expression). Iterative downsampling of regulons associated with clusters 3, 4, and cluster 4 (*SFRP4*+ subcluster, i.e., myofibroblasts), showing upregulated *IRF7*, *STAT1*, and *CREB3L1* regulons (**e**). For clustering panels, red = high, blue = low expression.).

identified *SMAD2* or *SMAD3* regulons, the canonical TFs associated with TGF-β activation. However, if we selected 984 differentially regulated genes between SSc *SFRP4*+ myofibroblasts, the remaining SSc *SFRP2*hi cells and the control *SFRP2*hi fibroblasts, SCENIC detected the SMAD3 regulon. This analysis showed upregulated regulons in a graded fashion, higher in *SFRP2*hi*PRSS23*+ subcluster 4 fibroblasts and highest in *SFRP4*+ myofibroblasts, again with some overlapping TFs seen in the previous analyses, including *STAT1, FOSL2, RUNX1,* and *FOXP1* (Supplementary Fig. 20). In this analysis, the SMAD3 regulon was shown upregulated, but its regulation was only associated with the transition between *SFRP2*hi*WIF1*+ (subcluster 3) to *SFRP2*hi-*WIF1−* (subcluster 4) fibroblasts and was actually decreased in *SFRP2*hi*SFRP4*+, myofibroblasts (Supplementary Fig. 20).

Since SCENIC networks are constructed from the same scRNA-seq dataset they are applied to, we also analyzed predicted TFs based on DoRothEA[55,56], which relies on independent TF-targeted gene interactions (regulon activity) curated from various resources, such as the literature, ChIP-seq peaks, TF-binding motifs, and gene expression inferred interactions. Based on the level of supporting evidence, DoRothEA computed regulons showed several TFs seen on the SCENIC analyses, most consistently *SMAD3, STAT1, FOSL2,* and *HIF1A* regulons, upregulated in *SFRP2*hi*WIF1−* (*PRSS23*+) fibroblasts in most interaction confidence levels, including level A, the level associated with the highest confidence interactions (Supplementary Fig. 21).

As previous studies have strongly implicated TGF-β in SSc pathogenesis[5,51], we further validated the role of *SMAD3*, its downstream TF, on SCENIC-predicted target genes in the transcriptome of SSc fibroblasts. To this end, we compared the genes included in the SCENIC, *SFRP2*hi*WIF1−* (*PRSS23*+), SMAD3 regulon with genes consistently upregulated by TGF-β1-, TGF-β2-, or TGF-β3-treated dermal fibroblasts (Supplementary data 10). TGF-β induced expression of *CHAC1*, a SCENIC-predicted downstream target of *SMAD3* was inhibited by SIS3, a specific inhibitor of Smad3 phosphorylation[57] in dermal fibroblasts from both control and SSc subjects (Supplementary Fig. 22).

Because *SMAD3* was expressed at low levels (in only a fraction of the cells), regulation of other *SMAD3* targets predicted by

SCENIC were difficult to assay, even by RT-PCR. Thus, to gain further insight into the role of *SMAD3*, we developed a SMAD3 activity index based on DoRothEA target A genes. Cells in cluster 4 (SSc *SFRP2*hi fibroblasts) and some cells also in cluster 3, 0, and 8 (dermal sheath) were found to express higher SMAD3 activity scores (Supplementary Fig. 23).

To further examine *SMAD3* activity, we created SMAD3 activity indices from experimentally determined RNA expression after *SMAD3* knockdown in myofibroblasts. *SMAD3* siRNA depressed *SMAD3* expression to 6.7% of nontargeting siRNA treatment (see Supplementary Data 11). These activity indices showed increased SMAD3 regulon activity in a more restrictive pattern, the highest activity was in cluster 4 (SSc *SFRP2*hi fibroblasts) and enhanced in the region of the *SFPR2*hi, *SFRP4*+ myofibroblasts, as well as in cluster 8 (dermal sheath cells, Supplementary Fig. 24).

## Discussion

We show here through bioinformatics and co-staining methods that myofibroblasts in SSc skin are a subpopulation of *SFRP2*hi-expressing fibroblasts. We have shown previously that these cells represent the most common fibroblast population in the skin, with a long narrow morphology that is similar to the morphology seen on staining SSc myofibroblasts with SMA[3]. Bioinformatics analyses show that *SFRP2*hi and myofibroblasts share closely related transcriptomes and pseudotime analysis indicates that SSc myofibroblasts derive from *SFRP2*hi*PRSS23*+*WIF1−* fibroblasts, an *SFRP2*hi fibroblast subpopulation. Our human SSc data is consistent with murine data, showing that *DPP4*-expressing fibroblasts in mice are profibrotic in wound healing[26]. However, *DPP4*, along with *SFRP2*, mark the largest population of fibroblasts in human dermis[27], and our scRNA-seq data provide more specific markers for this and related fibroblast subpopulations. The Rinkevich et al. cell lineage tracing study strongly supports the pseudotime analysis of our data, showing that the *SFRP2/DPP4* fibroblast subpopulation in healthy skin is the progenitor of fibrogenic fibroblasts in SSc skin, including both *SFRP2*hi*PRSS23*+*WIF1−* fibroblasts and myofibroblasts[26].

Our data show a global shift in fibroblast phenotypes in SSc skin. This includes increased expression of *PRSS23* and other genes by *SFRP2*hi fibroblasts, but also a shift within the

population of *APOE*[hi]/*CCL19*/*C7* fibroblasts, which show strikingly upregulated expression of a distinct series of genes like *CCL19* not upregulated in SSc *SFRP2*[hi] fibroblasts. These observations indicate that SSc is not a disease affecting only myofibroblasts. On the other hand, many genes, such as *TNC*, are regulated across different fibroblast subpopulations in SSc skin, suggesting that these different fibroblasts are being exposed to a common stimulus, such as Wnt or TGF-β.

Fibroblasts in SSc skin differentiate into myofibroblasts in two steps. The first step, a global shift of *SFRP2*[hi]*WIF1*+ fibroblasts to *SFRP2*[hi]*PRSS23*+*WIF1*− fibroblasts, is likely parallel to that described in Nazari et al., which was mimicked in vitro by inflammatory stimuli (TNFα and LTβ), but not TGF-β[36]. However, some of the key genes upregulated in this first step, such as *TNC* and *THBS1*, are known TGF-β-responsive genes, and both SCENIC and DoRothEA predicted *SMAD3* as regulating the transcriptome of cell in this step. Other data more strongly support TGF-β as driving the second step, transition of *SFRP2*[hi]*PRSS23*+*WIF1*− fibroblasts to myofibroblasts, as many of the genes upregulated in myofibroblasts are known TGF-β targets and are in a cluster of genes downregulated in the skin of SSc patients after treatment with anti-TGF-β/fresolimumab treatment of SSc patients, such as *THBS1*, *COMP*, *SERPINE1*, *COL10A1*, *CTGF*, and *MATN3* (ref. [51]). Data mapping genes from *SMAD3* knockdown experiments supported the role of *SMAD3* in both of these steps.

In contrast to work showing *DPP4* fibroblasts as profibrotic[26], other murine studies have shown that adipocytes[35], pericytes[34], and myeloid cells[58,59] or combinations of these in addition to resident fibroblasts[33] can act as myofibroblast progenitors. Lineage tracing experiments have elegantly shown that adipocytes at the interface with the dermis contribute to myofibroblasts found upon bleomycin-induced skin fibrosis[35]. Based on observations in patients with less severe myofibroblast infiltration, myofibroblasts appear first at the interface between subcutaneous fat and reticular dermis[3]. Notably, in vitro, TGF-β induces *SFRP2*, *TNC*, and *CTGF* expression by adipocytes differentiated in vitro from human adipose-derived progenitors, suggesting that adipocytes might differentiate into *SFRP2*[hi]*PRSS23*+*WIF1*− fibroblast, myofibroblast progenitors. Despite the proximity of SSc myofibroblasts to fat, we did not see any transcriptome relationship or overlap in specific markers between preadipocytes and myofibroblasts. Another study has shown that *PDGFRA*/*PDPN*, *ADAM12*-expressing perivascular cells are progenitors of myofibroblasts in murine skin and neural injury[34]. We found *ADAM12* expression highly induced in SSc myofibroblasts, these cells also expressing *PDGFRA* and *PDPN*. This contrasts to *ADAM12*-expressing perivascular progenitors, which downregulate *ADAM12* during myofibroblast differentiation. In addition, we did not see co-expression of pericyte markers to suggest myofibroblast differentiation from a pericyte progenitor, and pericyte populations did not significantly express *ADAM12* (see Supplementary Data 4). Although we cannot exclude the possibility that SSc myofibroblasts differentiate from a non-pericyte perivascular progenitor, our pseudotime analysis as well as SSc myofibroblast morphology and topological location indicates that they differentiate from *SFRP2*[hi] fibroblasts, cells that are distributed throughout normal dermis. Another recent scRNA-seq study identified myeloid cell markers *CD45* and *LYZ2* in a subcluster of wound myofibroblasts[60]. However, we did not observe expression of any myeloid marker genes in SSc *SFRP2*[hi]*PRSS23*+*WIF1*− or SSc myofibroblasts (Supplementary Data 8). In contrast to these cell types, in which we could find little transcriptome evidence for a progenitor relationship, we found multiple genes shared between dermal sheath cells and myofibroblasts, including high *ACTA2* expression, the gene encoding SMA[37]. Dermal sheath cells express other markers common to myofibroblasts, including *COL11A1*, and cluster proximal to myofibroblasts in UMAP plots. Despite these similarities, dermal sheath cells do not appear to be the direct progenitors of myofibroblasts in SSc skin, as each cell type expresses distinct sets of genes, and SSc fibroblasts are transcriptionally and topologically much more closely related to *SFRP2*[hi]*PRSS23*+*WIF1*− fibroblasts.

We also show increased proliferation of *SFRP2*[hi]*PRSS23*+*WIF1*− fibroblasts in SSc skin. Although, this low rate of proliferation is unlikely to account for the appearance of this cell type in SSc skin, it does suggest that a fibroblast growth factor contributes to the altered phenotype of these cells.

Several genes regulated in SSc myofibroblasts are reciprocally regulated compared to SM22 promoter tdTomato sorted wound myofibroblasts, as the wound heals and the fibroblasts lose SMA expression[61] (Supplementary Data 9): *TNC*, *SERPINE2*, *IGFBP3* (increased in SSc and early SMA+ wound myofibroblasts), and *WIF1* (decreased in SSc and late SMA+ wound myofibroblasts). Despite these parallels most gene expression changes seen in SSc myofibroblasts are distinct from those seen in wound myofibroblasts. This may have to do with the limitation of selecting myofibroblasts based on the SM22 promoter, as *TAGLN* (the target of the SM22 promoter) is also expressed by SMCs[62], and we see *TAGLN* (the target of the SM22 promoter) also highly expressed by pericytes and dermal sheath cells (not shown). Alternatively, wound and SSc myofibroblasts may represent different cell types and originate from different progenitors.

We originally described altered Wnt pathway gene expression in skin fibrosis, showing that Wnt2 and *SFRP4* mRNAs are strongly upregulated in the Tsk murine model of skin fibrosis, as well as in SSc skin biopsies[38]. Subsequently in a more comprehensive analysis of Wnt-related genes, we confirmed upregulated and correlated expression of *WNT2* and *SFRP4* gene expression in SSc skin[63]. We show here that the correlated upregulation of *WNT2* and *SFRP4* expression in SSc skin is most likely due to their co-regulation in SSc myofibroblasts. We also showed previously that SSc skin shows markedly decreased expression of *WIF1* (−7.88-fold), a soluble Wnt inhibitor[63]. Subsequent work by others confirmed downregulation of *WIF1* and increased Wnt activity[64]. Our data here show that downregulated *WIF1* is a marker for a global shift in the phenotype of *SFRP2*[hi]-expressing fibroblasts as they transition from *SFRP2*[hi]*WIF1*+ to *SFRP2*[hi]*PRSS23*+*WIF1*− fibroblasts. As we have previously shown that WIF1 expression in whole skin biopsies correlates strongly inversely with the MRSS[42], this supports the importance of this global shift in fibroblast phenotype that appears to precede the differentiation of these cells into myofibroblasts.

Several studies have implicated Wnts in fibrosis[65]. Wnt pathway activation increases both fibrillin matrix[63] and collagen expression. Wnt10b or β-catenin overexpression in mice leads to dermal fibrosis with increased expression of *COL1A1*, *COL1A2*, *CTGF*, and *ACTA2* mRNA in the skin[66,67]. Wnt3a blocks preadipocyte differentiation into adipocytes and stimulates their differentiation into myofibroblasts[64]. Other studies indicate that TGF-β mediates fibrosis by inhibiting *DKK1*, an endogenous Wnt inhibitor, leading to unrestrained profibrotic Wnt activity[68]. Although Wnt10b and *DKK1* upregulation and downregulation, respectively, have been identified by IHC[66,68], our microarray of whole skin shows *Wnt10b* expression decreased and *DKK1*, *DKK2*, and *DKK3* all increased in SSc compared to control skin (combined microarray data; R. Lafyatis). Thus, if indeed these Wnts are playing key roles in SSc pathogenesis, then there is a disconnect between mRNA and protein expression of these genes. This is not an unusual occurrence and indeed a significant limitation to gene expression analyses. However, we propose that altered expression of *WIF1*, *SFRP4*, and *WNT2*, all of whose

expression correlates highly with the MRSS, are more likely the key deregulated Wnts in SSc skin.

Regulon analysis implicated several unexpected TFs in regulating the transcriptome of SSc fibroblast differentiation, particularly *STAT1*, *FOSL2*, *RUNX1*, *IRF7*, *HIF1A*, *CREB3L1*, and *FOXP1*, as well as *SMAD3*. *IRF7* is upstream[69] and *STAT1* downstream[70,71] interferon signaling, previously implicated in SSc skin[41]. Polymorphisms in the *IRF7* and *HIF1A* genes are associated with SSc[72,73]; *IRF7* can bind *SMAD3*, and regulate fibrosis and profibrotic gene expression[74], while *HIF1A* has been implicated in mediating hypoxia-induced skin fibrosis[75]. Transgenic *FOSL2* overexpression leads to murine skin fibrosis, reproducing several features of SSc[76]. *FOSL2* is induced by TGF-β and regulates collagen production. Thus, several of the TFs predicted to regulate SSc fibroblast differentiation have been implicated in SSc skin fibrosis.

In conclusion, we identify the transcriptome of SSc myofibroblasts and show that *SFRP4* is an immunohistochemical marker for these cells. Further, our bioinformatics analyses indicate that myofibroblasts differentiate in a two-step process from *SFRP4/DPP4*-expressing normal fibroblast progenitors. These data also provide direct insights into previous studies of altered gene expression in SSc skin. We anticipate that applying scRNA-seq in clinical trial settings will enable far greater insights into the effects of therapeutics on the complex alterations of various cell types in SSc skin. We also expect that these observations will provide insights into myofibroblast origin and differentiation in other fibrotic diseases.

## Methods

**Study approval.** The University of Pittsburgh Medical Center Institutional Review Board (Pittsburgh, PA, USA) reviewed and approved the conduct of this study. Written informed consent was received from all participants prior to inclusion in the study.

**Single-cell RNA-sequencing.** A 3 mm skin biopsies were obtained from study subjects, digested enzymatically (Miltenyi Biotec Whole Skin Dissociation Kit, human) for 2 h and further dispersed using the Miltenyi gentleMACS Octo Dissociator. The resulting cell suspensions were filtered through 70 μm cell strainers twice and resuspended in phosphate-buffered saline (PBS) containing 0.04% BSA. Resulting cell suspensions were loaded into 10× Genomics Chromium instrument (Pleasanton, CA) for library preparation. V1 and V2 single-cell chemistries were used per manufacturer's protocol. Libraries were sequenced (~200 million reads/sample), using the Illumina NextSeq-500 platform. The sequencing reads were examined by quality metrics, transcripts mapped to reference human genome (GRCh38) and assigned to individual cells according to cell barcodes, using Cell Ranger (10× Genomics).

Data analysis was performed using R (version 3.6). Seurat 3.0 was used for data analysis, normalization of gene expression, and identification and visualization of cell populations[77,78]. Cell populations were identified based on gene markers and visualized by t-SNE[79] or UMAP[80] plots. We used AddModuleScore to calculate the average expression levels of each program (cluster) on a single-cell level, subtracted by the aggregated expression of control feature sets. Pathway analysis was performed with Gene Ontology Enrichment Analysis. Data presented were normalized between samples using SCTransform, which models technical noise using a regularized negative binomial regression model[81].

**Immunohistochemistry and Imaging.** Tyramide SuperBoost kit (Invitrogen, USA) was used to amplify signals in co-stained tissues, as per manufacturers protocol. Briefly, IF of formalin-fixed, paraffin-embedded human forearm skin biopsies were first, deparaffinized, and rehydrated for antibody staining. Slides were placed in citrate buffer pH 6, steamed for 20 min and cooled 20 min at room temperature for heat-induced antigen retrieval before washing in PBS. All primary antibodies were incubated overnight at 4° C. All poly-HRP secondary antibodies were used as per manufacturers protocol along with tyramide stock solution. Tyramides were incubated for 5 min each before neutralized with stop solution. Monoclonal mouse anti-SMA (1:1000; M0851; Clone14A; Dako, Denmark AS, Denmark) was labeled with Alexa Fluor 647 tyramide solution. In order to multiplex staining of slides with antibodies from the same species, slides were placed in citrate buffer pH 6, steamed for 20 min and cooled 20 min at room temperature for unbound antibody stripping before washing in PBS and proceeding with next antibody. Next, monoclonal mouse SFRP2 (1:250; MAB539; Millipore, USA) labeled with Alexa Fluor 488 tyramide solution was applied and washed, and then

polyclonal rabbit SFRP4 (1:500;153287-1-AP; Proteintech, USA) was labeled with Alexa Fluor 568.

For single staining polyclonal rabbit anti-CCL19 (1:500, ab221704, Abcam, USA) was labeled with Alexa Fluor 568 tyramide solution; monoclonal mouse anti-CRABP1 (1:500, MA3-813, C-1, Thermo Fisher, USA) was labeled with Alexa Fluor 488 tyramide solution; polyclonal rabbit anti-POSTN (1:250, ab14041, Abcam, USA) was labeled with Alexa Fluor 568 tyramide solution; and monoclonal mouse anti-SLPI (1:50, [31] ab17157, Abcam, USA) was labeled with Alexa Fluor 568 tyramide solution.

All slides were counterstained with nuclear stain Hoechst 33342 (Invitrogen, USA) and cover slipped with Pro-Long™ Diamond Antifade Mountant (P36961: Life Technologies, USA). Images were acquired using an Olympus Fluoview 1000 Confocal Scanning microscope.

**Transcription factor inference-SCENIC.** In order to better understand the TFs activating gene expression in SSc fibroblasts and myofibroblasts, we utilized SCENIC[54], a computational method for detecting gene regulatory networks. Embedded in this method is the identification of regulons, groups of genes identified by their co-expression with (GENIE3, ref. [82]), further selected by showing that genes in the regulon are enriched for TF *cis*-regulatory motifs. SCENIC then scores each cell for the level of gene expression by genes in each regulon, reported as AUC.

For the SCENIC analyses, we used only cells from V2 chemistries, four control and nine SSc samples. To begin clusters 1, 2, 3, and 4 were subsetted from the fibroblast dataset and all genes showing expression in at least one cell were analyzed. A second analysis was then carried out subsetting clusters 3 and 4, with cluster 4 further subsetted to delineate the SFRP4+ myofibroblast group. Again, all genes showing expression in at least one cell were included in the analysis. Alternatively, we filtered cells based on the workflow provided by Aerts lab[54], keeping genes that (1) with at least six UMI across all cells, and (2) detected in at least 1% of cells.

Finally, to focus on changes associated with SSc, a more restrictive gene list (984 genes) was compiled of (1) genes increased in *SFRP2*hi SSc cells (in clusters 3 and 4) compared to control *SFRP2*hi cells (in clusters 3 and 4, Bonferroni corrected Wilcoxon $p < 0.05$); and (2) genes increased in *SFRP2*hi*SFRP4*+ myofibroblasts compared to SSc *SFRP2*hi*SFRP4*− cells (in cluster 3 and 4, Bonferroni corrected Wilcoxon $p < 0.05$).

Using SCENIC we then analyzed the scRNA-seq expression matrices by GENIE3 to infer the co-expression network. GENIE3's output, a link list, included the potential regulators for each gene along with their weights, these weights representing the relevance the TF has in the prediction of the gene target.

**DoRothEA and VIPER.** We used VIPER in combination with DoRothEA to estimate TF activities from gene expression data[55]. DoRothEA contains 470,711 interactions, covering 1396 TFs targeting 20,238 genes, which rely on the independent TF-targeted gene interactions (regulon activity) curated from various resources, such as literature, ChIP-seq peaks, TF-binding motifs, and gene expression inferred interactions. Based on the number of supporting evidence that accompany each interaction, an interaction confidence level is assigned, ranging from A to E, with A being the highest confidence interactions and E the lowest. VIPER is a statistical method that used in combination with DoRothEA to estimate TF activities from scRNA-seq expression data[83].

We used TF target genes from DoRothEA, level A and from genes downregulated by siRNA to *SMAD3* to create *SMAD3* activity scores. Activity scores were also derived from *SMAD3* siRNA-treated myofibroblasts, filtered for absolute gene expression of nontargeting control siRNA >50 TPM, showing expression of *SMAD3* siRNA-treated cells of <0.8 of nontargeting control siRNA, and excluding genes showing expression <0.7 in *HRPT1* siRNA-treated cells compared to nontargeting control siRNA (415 genes); or for absolute gene expression in nontargeting control treated siRNA cells of >100 TPM, showing expression of *SMAD3* siRNA-treated cells of <0.7 of nontargeting control siRNA, and excluding genes showing expression <0.7 in *HRPT1* siRNA-treated cells compared to nontargeting control siRNA (74 genes).Using the Seurat AddModuleScore, we plotted these activity scores on scRNA-seq UMAP feature plots.

**Downsampling.** In order to test if the difference in cell numbers among clusters of 3 (*WIF1*+), 4 (*PRSS23*+), and 4 (*SFRP4*+; 673, 743, 73, respectively) would affect the SCENIC regulatory analysis, we used R function "sample" to randomly select 73 cells from cluster 3-*WIF1*+ and 4-*PRSS23*+, and then performing SCENIC analysis. We downsampled and performed SCENIC analyses 40 times with the resulting, different pools of cells.

**Pseudotime analysis.** Expression values were normalized in Monocle 3 (accounting for technical variation in RNA recovery, as well as sequencing depth) by estimating size factors for each cell and the dispersion function for genes[45,84]. Nonlinear dimensionality reduction was performed using UMAP. Cells were organized into trajectories by Monocle using reversed graph embedding (a machine learning strategy) to learn tree-like trajectories. Once a principal graph had been

learned, each cell was projected onto it, using SimplePPT, the default method in Monocle 3; it assumes that each trajectory is a tree that may have multiple roots.

**RNA velocity analysis.** Velocyto, a package for the analysis of expression dynamics in scRNA-seq data based on spliced and unspliced transcript reads, was used to estimate the time derivative of the gene expression state[46]. RNA velocities of cells in clusters 1, 3, and 4 were estimated using gene-relative model with $k$-nearest neighbor cell pooling ($k = 50$) based on top 100 differentially expressed genes from myofibroblast, cluster 3, and cluster 4. Velocity fields were projected into a UMAP-based embedding through SeuratWrappers in Seurat.

**Microarray analyses.** We combined and clustered microarray data from our previous biomarker study and clinical trials conducted by our center using the same Affymetrix U133A2.0 microarray chips[42,51–53]. Data were normalized using the MAS 5.0 algorithm, gene expression values were clustered using cluster 3.0 (ref. [85]). After filtering for genes showing differences of >100 across all samples, genes were mean centered, normalized, hierarchically clustered by complete linkage, and visualized using Java Treeview[86]. Hierarchical clusters (groups of genes referred to as signatures), corresponding to genes associated with the transition of healthy SFRP2hi fibroblasts into SSc SFRP2hi fibroblasts (PRSS23, TNC, and THBS1) and into myofibroblasts (CTGF, ADAM12, COL10A1, and MATN3) were analyzed, using the Seurat AddModuleScore function. The Seurat AddModuleScore function calculates the difference between the average expression levels of each gene set compared to random control genes at a single-cell level. These values were then plotted on $t$-SNE feature plots.

**Cell culture, and TGF-β, phospho-SMAD3, and siRNA analyses.** Early passage human dermal fibroblasts from SSc or healthy control skin were cultured in DMEM supplemented with 10% FBS after collagenase digestion. Fibroblasts were passaged at ~80% confluence, the following day placed in 0.1% serum and treated with TGF-β1, TGF-β2, or TGF-β3 (R&D Systems) or left untreated (control), or pretreated one hour with the Smad3 phosphorylation inhibitor SIS3 (CAS 1009104-85-1, Sigma Aldrich). After 16 h, RNA was prepared and analyzed by microarray Affymetrix U133A2.0 microarray chips as above, or cDNA prepared and analyzed by RT-PCR, using primers to CHAC1 Taqman FAM-MGB dye (Thermo Fisher #4331182) or SMAD3 Taqman FAM-MGB dye (Thermo Fisher #4453320); primer sequences are provided in Supplementary Data 12. Ct values for each treatment were normalized to 18S. Delta Ct for sample was then normalized to the control treatment and Fold change calculated.

For siRNA experiments, pulmonary myofibroblasts (passage 6) isolated from lung explants were used in knockdown experiments. Cells were passaged at 60% confluence, washed and then siRNAs targeting SMAD3, HRPT1 (control), or nontargeting control transfected 8 h, using Lullaby Transfection Buffer (OZ Biosciences; LL71000) in 200 μL OptiMEM with 5 μL 10 μM reconstituted dsRNAs (TriFECTa DsiRNA kit, cat#: hs.Ri.SMAD3.13, Integrated DNA Technology). After 48 h RNA was isolated from the cells with the RNeasy kit (Qiagen), quantified and quality checked in TapeStation High-Sensitivity RNA Screen Tape. cDNA libraries were synthesized and 25 million single-end reads sequenced per sample, using an Illumina High-Throughput Sequencer. FastQC reports were used to ensure quality data was entered into analysis. Alignment and gene counts were carried out using CLC Genomics version 20.0.3. Transcripts per kilobase million (TPM) were exported for samples and log fold changes were calculated with respect to the nontargeting controls.

**Statistics.** For tables examining differential gene expression between cells or groups of cells within clusters, cells were filtered out that were expressed in <10% of the cells showing upregulated expression. Comparisons of average numbers of cells in each fibroblast subpopulation were compared using Wilcoxon rank-sum test. Differential gene expression between healthy controls, and SSc was assessed using Seurat's implementation of the nonparametric Wilcoxon rank-sum test. A Bonferroni correction was applied to the results. Differences between the average proportions of cells in all control and SSc clusters were compared, using the chi square test. Individual differences between proportions of cells in each patient comparing SSc with controls cluster were calculated using Mann–Whitney. All statistical tests were two-sided.

**Reporting summary.** Further information on research design is available in the Nature Research Reporting Summary linked to this article.

## Data availability
All data supporting the findings from this study are available within the manuscript and the supplementary information. All scRNA-seq data including Gene cell UMI matrix and a BAM file containing aligned reads are available at the Gene Expression Omnibus: GSE138669. Source data are provided with this paper.

## Code availability
Code for data analyses is available and referenced in the text.

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

## Acknowledgements

We would like to acknowledge Dr. Simon Watkins and the University of Pittsburgh Center for Biological Imaging for technical assistance and use of imaging facilities. This research was supported in part by the University of Pittsburgh Center for Research Computing through the resources provided. This project used the University of Pittsburgh Health Sciences Sequencing Core at UPMC Children's Hospital Pittsburgh, Illumina sequencing.

## Author contributions

M.H., T.T., C.M., D.E.M., M.B., R.B., and A.P. contributed to designing research studies, conducting experiments, acquiring data, analyzing data, and writing the manuscript. W.C., M.J., and P.V.B. contributed to analyzing data, and writing the manuscript. R.D. contributed to designing research studies, conducting experiments, and writing the manuscript. R.L. contributed to designing research studies, analyzing data, and writing the manuscript.

## Competing interests

R.L. has served as a consultant for Pfizer, Bristol Myers Squibb, Boehringer-Ingleheim, Formation, Sanofi, Boehringer-Mannheim, Merck, and Genentech/Roche, and holds or recently had research grants from Corbus, Formation, Moderna, Regeneron, Pfizer, and Kiniksa. R.D. has worked as a consultant for Eicos Sciences Inc. The remaining authors declare no competing interests.
