## [Peer Review File · Nature Communications]

Reviewers' comments:

Reviewer #1 (Remarks to the Author):

Tabib and coworkers perform single-cell RNA sequencing on 12 skin samples obtained from patients with systemic sclerosis (SSc), compared to 10 skin samples from healthy donors. Main findings of the study are: (1) Confirming and expanding previous studies from this group and others, human skin contains several distinct fibroblast populations that are characterized by specific expression profiles (2) In contrast to other cell types in skin, fibroblast populations undergo global shift in their expression profile in SSc. (3) Dermal fibroblasts do not converge into one common activated phenotype but preserve heterogeneity in SSc. (4) Myofibroblasts in SScs arise from a distinct fibroblast progenitor population as shown by pseudotime analysis. (5) Myofibroblasts are not formed by transdifferentiation of pericytes or macrophages. (6) SSc myofibroblasts are characterized by specific transcription factor profiles.

This a technically extremely well and thoroughly performed study elucidating the progeny of activated fibroblasts in a human condition associated with fibrosis. In addition to confirmatory findings, the study makes numerous novel observations and creates data that will be of great importance to the field. However, although at a very high level, the study is mostly descriptive and does not contain any mechanistic approaches.

1. For instance, which of the myofibroblast-characteristic markers (e.g., SFRP4) or involved transcription factors could be used as targets in therapeutic approaches? How does interfering with the identified signaling pathways affect the myofibroblast marker profiles, if at all?

2. Verification of the mRNA profiles at the protein and tissue level is missing with the one exception of an immunofluorescence co-staining of SFRP4, SFRP2, and smooth muscle alpha actin. Normal skin controls are missing in this figure.

3. Along with the descriptive nature of the study, the purpose of the analysis not entirely clear. Was the aim to identify novel fibroblast/myofibroblast markers, targets, pathways and how will the findings be translated?

4. It is curious that bona fide myofibroblast markers are not tested for their expression in different clusters (e.g., Figures 3 and 4). The authors introduce smooth muscle alpha actin, collagen 1A1, TNC, CDH11, THY1 as frequently used myofibroblast markers but it is not investigated in tSNE or UMAP

analysis which fibroblast populations are characterized. Likewise, the authors do not compare their fibroblast clusters (and specific markers) with clusters that have been produced by other groups in different fibrotic disease context or lineage-tracing studies.

Reviewer #2 (Remarks to the Author):

This study is an extension of previous study on single cell RNA seq analysis of healthy normal skin (Tabib T., et al JID 2018). In this manuscript, the authors performed additional single cell RNA-seq analysis on skin biopsies of 12 patients with SSc and 10 control samples, 4 of which were added from the previous study. Based on clustering analysis, the authors reveal 10 fibroblast cell types in SSc that are also represented in control samples. Fibroblasts expressing high levels of SFRP2 represent the major population of dermal fibroblasts where a sub cluster positive for WIF1 is depleted while cluster expressing PRSS23 is increased in SSc skin fibroblasts. The sub cluster expressing SFRP2(hi) and PRSS23 correlate with the severity of SSC disease and authors apply Monocle to establish cell trajectory and suggest that SFRP(hi)PRSS23+WIF1- fibroblasts are the immediate progenitors of myofibroblasts, where minor population of them are highly proliferative in SSc skin. The authors further compare the data with bulk mRNA expression showing PRSS23 correlated highly with the MRSS and authors use SCENIC tool to infer transcription factor networks associated in SSC transition from control fibroblasts. The manuscript is a descriptive paper revealing single cell RNA expression profiling of human control and SSC skin.

Major concerns:

1. On page 8, the authors claim that SSc patients showed a lower proportion of fibroblasts compared to controls with a relatively modest increase in the proportion of keratinocytes. However, based on Figure 1 panel A and C, the number of keratinocytes (clusters 0, 1, 4, 11) are mainly from SSC samples - very little from control. Please explain the discrepancy (e.g. was the epidermis removed during the sample preparation?). It will also be helpful to quantify these cell populations and show in terms of pie or column charts to demonstrate 'modest' increase as stated by the authors. Similarly on page 11, it would be helpful to quantify the 'global' changes in fibroblast population and illustrate them as a figure with statistics to demonstrate significant increase in reticular dermis population but not DP and DS fibroblasts. For instance, Fisher's exact test to show increase in proportion of cell population may suffice.
2. CCL19 subpopulation in cluster 0 in figure 2C seems significant but the authors do not address this cell type/state in much detail.

3. On Page 12, referring to figure 5, the authors claim that PRSS23, THBS1, TNFSF18, CTGF and FNDC1 are 'broadly' expressed in most cells in this cluster. Upon close examination of the figure, PRSS23, THBS1 and possibly CTGF exhibit similar expression patterns whereas TNFF18, FNDC1 show distinct patterns similar to SFRP4, COL10A1, MATN3. Please explain how these genes were selected and based on what objective criteria?

4. Figure 4B, immuno-staining images are not convincing. The DAPI, which stains the nucleus, seems elongated and blurred. The zoomed image could also be selective and not representative of the whole tissue. Please provide a wider view of the skin to show a greater context to the area of interest. For instance, including the barrier between the dermis and epidermis. As a control, it will also be helpful to show the stainings (especially the negative expression of SFRP4 and SFRP2) in healthy donors or in not so deep part of the dermis.

5. The authors suggest the directionality of the pseudo time (Figure 5) based on a subjective notion that SSc samples are found in cluster 4. Showing directionality using tools such as RNAVelocity would objectively support the author's claim that cells are transitioning from cluster 1, 3, and 4. In the same token, Figure 5 only highlights genes that are up-regulated across the pseudo time. Showing different trajectory patterns, ie. down regulation of genes that are associated with progenitor cells would also support the notion that the assumed directionality is correct.

6. A minor cluster (sub cluster 9) to be associated with proliferation is interesting but questionable. Single cell profiling platforms can easily produce artifacts or doublets with high expression of cell profiling markers. Since the numbers are low (39 in total), using flow cytometry methods to detect KI67 or other cell proliferative markers and myofibroblasts markers will be necessary.

7. How microarray and single cell gene sets were analyzed is completely missing in the methods section. Please clearly explain how the gene sets were selected and which statistics were applied to show enrichment of gene sets in bulk microarray data. Please define what authors mean by "signature".

8. The approach in which genes were selected and analyzed for SCENIC analysis is unclear and text is also confusing. Please provide a clearer explanation of gene sets used and which background was set to derive statistical significance. Few words are repeated in multiple occasions in the last paragraph of the results sections. Editing is advised.

9. In some analysis, UMAP clusters are used (e.g. MONOCLE) and in another analysis, TSNE clusters are used (e.g. SCENIC). Please use single definition of clusters across all analysis.

10. The claim that SFRP4 is a "robust immunohistochemical marker" for myofibroblasts seems far fetched based on single staining analysis. Demonstrating specificity across different tissue conditions (including healthy controls) and wider resolution to cover different areas of the skin will help to determine specificity and the robustness of the marker.

11. The claim that myofibroblasts differentiate in a "two-step" process from SFRP4/DPP4 expressing normal fibroblasts progenitors is also under substantiated. Rudimentary analysis based on SCENIC with arbitrarily selected genesets to infer TF regulons hardly proves that these progenitors are undergoing two distinct steps towards myofibroblasts. Either lineage tracing and/or ChIP seq analysis to demonstrate binding of regulations will strong support the claim made by the authors.

12. Surprisingly, most of CD45 immune cells are not detected. If they were removed during cell preparation, the authors should clearly state how they were removed in the methods section. If not removed, the data shows under-representation in the skin. We expect SSc patients to harbor larger number of CD45+ leukocytes and they will play a critical role in skin microenvironment and influence gene expression - how they contribute to the overall reactivity of myfibroblasts is missing in this manuscript.

Minor points:

1. The top section on page 12 seems redundant to the bottom section of page 11. Top section on page 12 also seems to skip figure 3. Similarly, Section: "Myofibroblasts show a discrete transcriptome" on page 13 is redundant. Similar claims were already made in page 12.
2. All Supplementary figures are missing. I was not able to access them.
3. There are too many UMAPS to juggle back and fourth to follow the results from the manuscript. Showing a single UMAP with SFRP2, PRSS23, SFRP4 and WIF1 expression will greatly help to understand the flow and intent of the authors. And move other parts into supplementary.
4. Figure 7c, show figure labels
5. Please highlight UMAP with which V1 and V2 kits was used (in the supplementary).
6. What is a "link list" in page 25. Rank list?

Manuscript # NCOMMS-19-34513-T

Title Myofibroblast transcriptome indicates SFRP2+ fibroblast progenitors in systemic sclerosis skin

Point-by-point response to reviewer comments.

Reviewers' comments:

Reviewer #1 (Remarks to the Author):

Tabib and coworkers perform single-cell RNA sequencing on 12 skin samples obtained from patients with systemic sclerosis (SSc), compared to 10 skin samples from healthy donors. Main findings of the study are: (1) Confirming and expanding previous studies from this group and others, human skin contains several distinct fibroblast populations that are characterized by specific expression profiles (2) In contrast to other cell types in skin, fibroblast populations undergo global shift in their expression profile in SSc. (3) Dermal fibroblasts do not converge into one common activated phenotype but preserve heterogeneity in SSc. (4) Myofibroblasts in SScs arise from a distinct fibroblast progenitor population as shown by pseudotime analysis. (5) Myofibroblasts are not formed by transdifferentiation of pericytes or macrophages. (6) SSc myofibroblasts are characterized by specific transcription factor profiles.

This a technically extremely well and thoroughly performed study elucidating the progeny of activated fibroblasts in a human condition associated with fibrosis. In addition to confirmatory findings, the study makes numerous novel observations and creates data that will be of great importance to the field. However, although at a very high level, the study is mostly descriptive and does not contain any mechanistic approaches.

1. For instance, which of the myofibroblast-characteristic markers (e.g., SFRP4) or involved transcription factors could be used as targets in therapeutic approaches? How does interfering with the identified signaling pathways affect the myofibroblast marker profiles, if at all?

We agree these are important questions to address in future work. We feel they are beyond the scope of the current manuscript.

2. Verification of the mRNA profiles at the protein and tissue level is missing with the one exception of an immunofluorescence co-staining of SFRP4, SFRP2, and smooth muscle alpha actin. Normal skin controls are missing in this figure.

We have added staining of CCL19, CRABP1, POSTN and SLPI, identifying other fibroblast populations in control and SSc skin, complementing our previous studies. We have stained SFRP2 recently in normal skin (J Invest Dermatol. 2018; 138(4):802-10). We previously stained SFRP4 in normal and systemic sclerosis skin in a cohort of 12 healthy and 19 systemic sclerosis skin samples (J Invest Dermatol. 2008; 128(4):871-81). Thus, we don't feel that adding normal staining of these antigens adds to the current manuscript. The additional immunofluorescent staining based on available antibodies to additional cell subsets. Most significantly staining healthy and SSc skin, showing that CRABP1 stains cells in human dermal papilla as shown previously in murine studies and Perisostin stains in the papillary dermis Figure S3C. Thus, this fibroblast subpopulation (subcluster #2 in Figure 2) represents a cell population, previously not identified in human skin as papillary fibroblasts. Further based on the work of Dr. Watt's group, (J Invest Dermatol. 2018; 138(4):811-25), we identify markers of papillary fibroblasts in fibroblast subcluster 3 (Figure 2). We have additionally added dot plot figures making it easier to follow the different cell types in normal skin (Figure S3B) and combined SSc and normal skin (Figure 3A). With these additional studies we have characterized most of the fibroblast subpopulations in skin.

3. Along with the descriptive nature of the study, the purpose of the analysis not entirely clear. Was the aim to identify novel fibroblast/myofibroblast markers, targets, pathways and how will the findings be translated?

The scRNA-seq data advances our understanding of SSc myofibroblast biology in several areas. Defining the transcriptome-phenotype of myofibroblasts, previously only defined by staining with smooth muscle active provides a first critical step in understanding their biology. In addition, the data clarify an important point regarding myofibroblast origins, i.e. that they derive from fibroblasts and not myeloid, adipocyte, pericyte or epithelial progenitors. The bioinformatics analysis provides a map for identifying the transcription factors that regulate myofibroblast differentiation. The data support a two-step mode for their differentiation and show that type I collagen is upregulated in the first step.

4. It is curious that bona fide myofibroblast markers are not tested for their expression in different clusters (e.g., Figures 3 and 4). The authors introduce smooth muscle alpha actin, collagen 1A1, TNC, CDH11, THY1 as frequently used myofibroblast markers but it is not investigated in tSNE or UMAP analysis which fibroblast populations are characterized. Likewise, the authors do not compare their fibroblast clusters (and specific markers) with clusters that have been produced by other groups in different fibrotic disease context or lineage-tracing studies.

Since our original submission dot plots have emerged as a more compact visually informative way to show expression of genes in different clusters. Thus, we have added a figure (Figure 3A), showing more easily accessible information about the markers of the different clusters, addressing this concern and other concerns below. We have added the genes suggested though until the work we have shown here, I do not believe that bona fide myofibroblast markers in SSc skin had been clearly defined. We have also added ADAM12 a proposed marker for pericyte derived myofibroblasts (Nature medicine. 2012; 18(8): 1262-70), and DPP4 a proposed marker of activated fibroblasts in SSc skin (Arthritis & rheumatology. 2020; 72(1): 137-49). Our data show that ADAM12 is a good marker for myofibroblasts, but not due to its expression on pericytes and that DPP4 is not a good marker of the myofibroblast differentiation but rather marks the fibroblasts in normal skin destined to become myofibroblasts.

Reviewer #2 (Remarks to the Author):

This study is an extension of previous study on single cell RNA seq analysis of healthy normal skin (Tabib T., et al JID 2018). In this manuscript, the authors performed additional single cell RNA-seq analysis on skin biopsies of 12 patients with SSc and 10 control samples, 4 of which were added from the previous study. Based on clustering analysis, the authors reveal 10 fibroblast cell types in SSc that are also represented in control samples. Fibroblasts expressing high levels of SFRP2 represent the major population of dermal fibroblasts where a sub cluster positive for WIF1 is depleted while cluster expressing PRSS23 is increased in SSc skin fibroblasts. The sub cluster expressing SFRP2(hi) and PRSS23 correlate with the severity of SSC disease and authors apply Monocle to establish cell trajectory and suggest that SFRP(hi)PRSS23+WIF1- fibroblasts are the immediate progenitors of myofibroblasts, where minor population of them are highly proliferative in SSc skin. The authors further compare the data with bulk mRNA expression showing PRSS23 correlated highly with the MRSS and authors use SCENIC tool to infer transcription factor networks associated in SSC transition from control fibroblasts. The manuscript is a descriptive paper revealing single cell RNA expression profiling of human control and SSC skin.

Major concerns:

1. On page 8, the authors claim that SSc patients showed a lower proportion of fibroblasts compared to controls with a relatively modest increase in the proportion of keratinocytes. However, based on Figure 1 panel A and C, the number of keratinocytes (clusters 0, 1, 4, 11) are mainly from SSC samples - very little from control. Please explain the discrepancy (e.g. was the epidermis removed during the sample preparation?). It will also be helpful to quantify these cell populations and show in terms of pie or column charts to demonstrate 'modest' increase as stated by the authors. Similarly, on page 11, it would be helpful to quantify the 'global' changes in fibroblast population and illustrate them as a figure with statistics to demonstrate significant increase in reticular dermis population but not DP and DS fibroblasts. For instance, Fisher's exact test to show increase in proportion of cell population may suffice.

We have added these data as suggested. We have added a column chart to show the different proportions of cells in Figure 1D. We did not remove the epidermis during preparation of the samples.

The difficulty with seeing the proportions of each cell type clearly in Figure 1C is because there are more SSc total cells (from more patients n=12) than control cells (n=10), and the blue "SSc" dots tend to cover up the pink "control" dots in 1C. We used a Chi square calculation to show that the shift in fibroblast populations is statistically significant ($p < 0.001$), and indicated the populations showing a change in proportion from Control to SSc skin. The data show that the only keratinocyte population showing a statistically significant change are the KRT6A population, expanding in SSc, the other populations showing only trends toward greater numbers in SSc. Changes in fibroblasts proportions are better described in the following figure focused on those cell types. For this reason, we have removed the sentence, "SSc patients showed a lower proportion of fibroblasts compared to controls with a relatively modest increase in the proportion of keratinocytes, whereas other cell types were balanced across control and SSc (not shown)."

We had previously shown the differential cell proportions in the fibroblast clusters (Figure 3B) We have added statistical analysis (Chi square and statistical comparisons between cell clusters).

2. CCL19 subpopulation in cluster 0 in figure 2C seems significant but the authors do not address this cell type/state in much detail.

Yes, we agree this cluster is potentially important, in part because it shows changes are occurring in multiple fibroblast populations. In the new dot plot Figure 3B, we have included CCL19 and APOE marker genes of the CCL19 population. Although there appear more CCL19+ from SSc cells than control fibroblasts, they were not statistically increased. They do cluster separately from the control cells in the cluster and express increased CCL19, indicating that they are altered in SSc. I have added this to the text. As with all the fibroblast populations, we have included a Table showing differentially expressed genes by these fibroblasts.

3. On Page 12, referring to figure 5, the authors claim that PRSS23, THBS1, TNFSF18, CTGF and FNDC1 are 'broadly' expressed in most cells in this cluster. Upon close examination of the figure, PRSS23, THBS1 and possibly CTGF exhibit similar expression patterns whereas TNFSF18, FNDC1 show distinct patterns similar to SFRP4, COL10A1, MATN3. Please explain how these genes were selected and based on what objective criteria?

Genes were selected by examining the top differentially expressed genes defining the different clusters. Yes, the reviewer is absolutely correct, and this is shown quite clearly on the new dotplot figure we have added, (Figure 3A). We have also added COL1A1 and TNC to that dotplot, these genes expressed in the broad pattern of PRSS23 and THBS1, whereas CTGF, ADAM12, TNFSF18, FNDC1, COL10A1 and MATN3 are expressed primarily by the myofibroblast subpopulations. I have changed this in the text

4. Figure 4B, immuno-staining images are not convincing. The DAPI, which stains the nucleus, seems elongated and blurred. The zoomed image could also be selective and not representative of the whole tissue. Please provide a wider view of the skin to show a greater context to the area of interest. For instance, including the barrier between the dermis and epidermis. As a control, it will also be helpful to show the stainings (especially the negative expression of SFRP4 and SFRP2) in healthy donors or in not so deep part of the dermis.

We have previously published images that show staining for SFRP4 in 12 healthy control and 19 SSc skin samples examining both superficial and deep dermis staining, and providing a statistical analysis comparing of SFRP4 expression and IHC staining. (J Invest Dermatol. 2008; 128(4):871-81). More recently we have shown that SRRP2 stains a broad, common subset of fibroblasts in normal dermis (J Invest Dermatol. 2018; 138(4):802-10). So, control skin staining is shown in these previous publications and would seem duplicative here. The elongated nuclei of these cells are consistent with what we have shown previously for this subset of fibroblasts. The nuclei are, as shown, elongated. This is an accurate representation of their morphology, again described in detail in J Invest Dermatol. 2018; 138(4):802-10. The point of Figure 4B is to show that SFRP4 staining cells in the deep dermis co-stain with, smooth muscle actin (i.e. ACTA2), and SFRP2 as shown by scRNA-seq at the level of

mRNA expression, showing that these triple, positive cells are myofibroblasts and closely related to normal SFRP2 fibroblasts.

5. The authors suggest the directionality of the pseudo time (Figure 5) based on a subjective notion that SSc samples are found in cluster 4. Showing directionality using tools such as RNAVelocity would objectively support the author's claim that cells are transitioning from cluster 1, 3, and 4. In the same token, Figure 5 only highlights genes that are up-regulated across the pseudo time. Showing different trajectory patterns, i.e. down regulation of genes that are associated with progenitor cells would also support the notion that the assumed directionality is correct.

As the reviewer suggests, we have added WIF1, which goes down in expression and cells differentiate from fibroblasts to myofibroblasts. We have shown previously that expression of this gene correlates negatively with the MRSS (Arthritis Rheumatol 67, 3004-3015; 2015). We have simplified this figure (Figure 5) so that the change in pseudotime of a restricted number of marker genes is more easily seen, placing the previous figure in the supplemental figures (Figure S11). The claim in differentiation from healthy fibroblasts to myofibroblasts in the pseudotime analysis is based on the knowledge that SSc patients start with normal skin and thus normal fibroblasts. There is significant body of data supporting this assertion. So, there is logically a movement from normal healthy to diseased fibroblast transition and transcription. As suggested, we have also added a Velocity analysis, supporting the direction of cell differentiation to myofibroblasts (Figure S12).

6. A minor cluster (sub cluster 9) to be associated with proliferation is interesting but questionable. Single cell profiling platforms can easily produce artifacts or doublets with high expression of cell profiling markers. Since the numbers are low (39 in total), using flow cytometry methods to detect KI67 or other cell proliferative markers and myofibroblasts markers will be necessary.

Given the very low numbers of dividing cells and the low numbers of cells digestible from scleroderma skin (4000-8000 cells), it is not practical to expect we would be able to see these cells by flow cytometry. While doublets can be a troubling artifact, in this case a doublet would not be expected to show the characteristic of a proliferating cell while also consistently showing another characteristic gene expression of a myofibroblast and no other cell type. If proliferating cells were forming doublets (which would be surprising as this is a stochastic event, typically seen in ~1% of cells) with other cell types then we would expect a random distribution with (markers of) other cell types.

7. How microarray and single cell gene sets were analyzed is completely missing in the methods section. Please clearly explain how the gene sets were selected and which statistics were applied to show enrichment of gene sets in bulk microarray data. Please define what authors mean by "signature".

We describe our analysis of single cell datasets in the Methods section "Single cell RNA-sequencing". Some of the methods for the microarray analysis were described in the Figure legend but we amplify in a new section in the Methods. We have added the following to the Methods. "Microarray Analysis. We combined and clustered microarray data from our previous biomarker study and clinical trials conducted by our center using the same Affymetrix U133A2.0 microarray chips {Rice, 2015 #200; Rice, 2015 #202; Lafyatis, 2017 #138; Mantero, 2018 #150}. Data were normalized using the MAS 5.0 algorithm, gene expression values were clustered using Cluster 3.0 {Eisen, 1998 #339}. After filtering for genes showing differences of greater than 100 across all samples, genes were mean centered, normalized, hierarchically clustered by complete linkage, and visualized using Java Treeview {Saldanha, 2004 #291}. Hierarchical clusters (groups of genes referred to as signatures), corresponding to genes associated with the transition of healthy SFRP2+ fibroblasts into SSc SFRP2+ fibroblasts (PRSS23, TNC, THBS1) and into myofibroblasts (CTGF, ADAM12, COL10A1 and MATN3) were analyzed, using the Seurat AddModuleScore function. The Seurat AddModuleScore function calculates the difference between the average expression levels of each gene set compared to random control genes at a single cell level. These values were plotted on t-SNE feature plots."

We do not make any statistical claims about this analysis. However, we have shown in previous publications that expression of several of these genes (THBS1, COMP, ADAM12, and CTGF) correlate highly and statistically significantly with the MRSS (Arthritis & rheumatology).

2015;67(11):3004-15), so the observation that these genes cluster together in bulk RNA-seq analysis and their co-expression in our single cell RNA-seq dataset in the transition of healthy SFRP2+ fibroblasts to SSc SFRP2+ fibroblasts and myofibroblasts is consistent with the roles of these SSc fibroblast populations in driving clinical disease. I have added this to the text.

We use "signature" rather than "cluster" for the cluster of genes from the hierarchical clustering used for the AddModuleScore analysis to help distinguish these from the clusters and subclusters referred to on the t-SNE and UMAP plots. I felt that using the word "signature" would help distinguish this analysis.

8. The approach in which genes were selected and analyzed for SCENIC analysis is unclear and text is also confusing. Please provide a clearer explanation of gene sets used and which background was set to derive statistical significance. Few words are repeated in multiple occasions in the last paragraph of the results sections. Editing is advised.

We apologize for not being clearer about our methods here. I have added the following to the METHODS section. "For the SCENIC analyses, we used only cells from V2 chemistries, 4 control and 9 SSc samples. To begin clusters 1, 2, 3, and 4 were subsetted from the fibroblast dataset and all genes showing expression in at least one cell were analyzed. A second analysis was then carried out subsetting clusters 3 and 4, with cluster 4 further subsetted to delineate the SFRP4+ myofibroblast group (Figure S13A). Again, all genes showing expression in at least one cell were included in the analysis. Finally, to focus on changes associated with SSc, a more restrictive gene list (984 genes) was compiled of: 1) genes increased in SFRP2+ SSc cells (in clusters 3 and 4) compared to control SFRP2+ cells (in clusters 3 and 4, Bonferroni corrected Wilcoxon $p < 0.05$); and 2) genes increased in SFRP2+SFRP4+ myofibroblasts compared to SSc SFRP2+SFRP4- cells (in cluster 3 and 4, Bonferroni corrected Wilcoxon $p < 0.05$)." We have edited the paragraph as suggested

9. In some analysis, UMAP clusters are used (e.g. MONOCLE) and in another analysis, TSNE clusters are used (e.g. SCENIC). Please use single definition of clusters across all analysis.

I understand this is confusing, but the clusters we analyze in MONOCLE (cluster 1, 3 and 4) and SCENIC (clusters 1-4), are the same clusters (by number) as in the UMAP plot. It just happens that these are the four (out of 10 clusters) in the UMAP (Figure 2A) most highly expressing SFRP2. To provide more clarity, we have changed the labeling of these clusters so that they are consistently labeled across the figures, using selectively expressed marker genes.

10. The claim that SFRP4 is a "robust immunohistochemical marker" for myofibroblasts seems far fetched based on single staining analysis. Demonstrating specificity across different tissue conditions (including healthy controls) and wider resolution to cover different areas of the skin will help to determine specificity and the robustness of the marker.

We several years ago we described SFRP4 staining in both superficial and deep dermis (*J Invest Dermatol.* 2008; 128(4):871-81). In that manuscript we show the staining in the superficial and deep dermis in control and SSc skin. There is a population of fibroblasts in the papillary dermis that also stain SFRP4 in normal skin as seen in that paper and also in our scRNA-seq data. So, we agree robust is too strong a term and rather the transcriptome, representing the multiple myofibroblast genes are the robust markers I have removed this descriptor from the text.

11. The claim that myofibroblasts differentiate in a "two-step" process from SFRP4/DPP4 expressing normal fibroblasts progenitors is also under substantiated. Rudimentary analysis based on SCENIC with arbitrarily selected genesets to infer TF regulons hardly proves that these progenitors are undergoing two distinct steps towards myofibroblasts. Either lineage tracing and/or ChIP seq analysis to demonstrate binding of regulations will strong support the claim made by the authors.

The paradigm for two-step differentiation is based on the alterations in gene expression rather than the SCENIC data. The dot plot added to Figure 3A shows this more clearly. Lineage tracing and chip-

seq are not possible using primary human fibroblasts, which we know from other work change their phenotype within a couple days after placing in tissue culture. The advantage of these scRNA-seq experiments is that they capture the transcriptome of the cells within a couple hours of their excision from skin biopsies.

12. Surprisingly, most of CD45 immune cells are not detected. If they were removed during cell preparation, the authors should clearly state how they were removed in the methods section. If not removed, the data shows under-representation in the skin. We expect SSc patients to harbor larger number of CD45+ leukocytes and they will play a critical role in skin microenvironment and influence gene expression - how they contribute to the overall reactivity of myofibroblasts is missing in this manuscript.

I'm not sure I understand why the reviewer states that most of the CD45 immune cells are not detected. We show these cells in clusters 7 (NK/T) and 9 (macrophages) in Figure 1. CD45 immune cells actually represent a relatively small proportion of the cells in SSc skin compared to many other pathologies, such as psoriasis or atopic dermatitis. Because of the complexity of the fibroblast analysis and of the myeloid cell populations in skin, we have prepared a separate manuscript describing myeloid populations in SSc skin. This is too large a topic to include in this manuscript.

Minor points:

1. The top section on page 12 seems redundant to the bottom section of page 11. Top section on page 12 also seems to skip figure 3. Similarly, Section: "Myofibroblasts show a discrete transcriptome" on page 13 is redundant. Similar claims were already made in page 12.

Although there is some overlap between these analyses, the paragraph entitled "Myofibroblasts show a discrete transcriptome" focuses on the transcriptome of myofibroblasts, a critical outcome of the manuscript.

2. All Supplementary figures are missing. I was not able to access them.

I'm not sure why these were not available. They will be again submitted with additions as noted above. The lack of availability of these Figures doubtless made it hard to follow parts of the manuscript.

3. There are too many UMAPs to juggle back and forth to follow the results from the manuscript. Showing a single UMAP with SFRP2, PRSS23, SFRP4 and WIF1 expression will greatly help to understand the flow and intent of the authors. And move other parts into supplementary.

We have reduced the number of feature plots in the main Figure section and shifted these to the Supplemental Figures, but also added and condensed much of this data to a new dot plot figure (Figure 3A), which allows easier, more condensed viewing of the different population.

4. Figure 7c, show figure labels

This has been added

5. Please highlight UMAP with which V1 and V2 kits was used (in the supplementary).

We have added this as Figure S5B.

6. What is a "link list" in page 25. Rank list?

The link list is the output of GENIE3 that tells us each gene's potential regulators and an associated weight based on the input data set. It lets us know, through the weight, how relevant a transcription factor/regulator is in relation to its target. The file has 3 columns with TF (transcription factor), Target (for the target gene), and weight. We create the transcription factor modules from this list.

REVIEWER COMMENTS

Reviewer #1 (Remarks to the Author):

The authors have not been dramatically responsive but I consider the work sufficiently novel and impactful to be published in Nat Comm.

Reviewer #2 (Remarks to the Author):

1. The authors addressed questions that were raised, however, the reviewer is still concerned regarding the statistics used to predict transcription factors in the SCENIC package and drawing general conclusions based on this.

1. Drop outs are a common problem of single cell RNA-seq thus how the data is filtered will dramatically influence the outcome of the analysis. As the authors noted, identification of specific motif can only be found when the number of input genes are reduced. In such case how do you adjust the background for statistical analysis, ie. is the same background applied for transcriptome and restricted set of genes? Please define background for each comparison and provide explanation how the enrichment is calculated in respect to the background. It should be noted that, while DoRothEA and D-AUCell rely on independent regulons, the SCENIC networks are constructed from the same dataset they are applied to. This poses the risk of overfitting. Please consider other tools for confirmation.

2. All genes showing expression in at least one cell seems very permissive as SCENIC and other regulatory analysis tools are easily influenced by the gene input. Please try with more robust cutoff and show significance of this analysis. Furthermore, the number of cells in each sub cluster are dramatically different. Unequal variance of cell number will influence the statistics to derive meaningful motifs. Please try using similar cell numbers (e.g. downsampling) and see if the conclusions are the same.

3. Why does SMAD3 only come up only when restricted genes are used as input? Again, this may be easily influenced by the input number and how the background is defined. To ensure SMAD3 is not an artifact, functional validation in culture fibroblasts or CHIP-seq validation (e.g. Cut & Tag) from tissue samples would substantially prove this.

2. The immune staining is still not convincing. The antibodies used in past studies differ from the one that is presented in this manuscript. Therefore a proper control is necessary.

In this paper:

polyclonal rabbit SFRP4 (1:500;153287-1-AP; Proteintech, USA)

In J Invest Dermatol. 2008;128(4):871-81,

rabbit polyclonal affinity-purified anti-SFRP4 antibody (1:70 dilution; kindly provided by Dr R. Friis)

In J Invest Dermatol. 2018;138(4):802-10

No SFRP4 was tested.

3. There is a big discrepancy between the single cell RNA-seq data where SFRP4 is a small percentage of SFRP2, the immune-staining seem to suggest nearly 100% overlap between the two markers. This either indicates problems with the staining background or single cell data is not representative of cell population. Please clarify.

4. Figure 8 panel E and F should be swapped.

REVIEWER COMMENTS

Reviewer #1 (Remarks to the Author):

The authors have not been dramatically responsive but I consider the work sufficiently novel and impactful to be published in Nat Comm.

We thank the reviewer for agreeing that the manuscript is novel and impactful.

Reviewer #2 (Remarks to the Author):

1. The authors addressed questions that were raised, however, the reviewer is still concerned regarding the statistics used to predict transcription factors in the SCENIC package and drawing general conclusions based on this.

1. Drop outs are a common problem of single cell RNA-seq thus how the data is filtered will dramatically influence the outcome of the analysis. As the authors noted, identification of specific motif can only be found when the number of input genes are reduced. In such case how do you adjust the background for statistical analysis, ie. is the same background applied for transcriptome and restricted set of genes?

Response: Our the initial SCENIC analyses was carried out using all genes expressed by at least one cell (so these aren't filtered, since we are only removing undetected genes). From our previous response, "For the SCENIC analyses, we used only cells from V2 chemistries, 4 control and 9 SSc samples. To begin clusters 1, 2, 3, and 4 were subsetted from the fibroblast dataset and all genes showing expression in at least one cell were analyzed. A second analysis was then carried out subsetting clusters 3 and 4, with cluster 4 further subsetted to delineate the SFRP4+ myofibroblast group (Figure S13A). Again, all genes showing expression in at least one cell were included in the analysis. Finally, to focus on changes associated with SSc, a more restrictive gene list (984 genes) was compiled of: 1) genes increased in SFRP2+ SSc cells (in clusters 3 and 4) compared to control SFRP2+ cells (in clusters 3 and 4, Bonferroni corrected Wilcoxon $p < 0.05$); and 2) genes increased in SFRP2+SFRP4+ myofibroblasts compared to SSc SFRP2+SFRP4- cells (in cluster 3 and 4, Bonferroni corrected Wilcoxon $p < 0.05$)." The validity of the SCENIC analysis is addressed further below, applying DoRothEA in a parallel analysis.

Please define background for each comparison and provide explanation how the enrichment is calculated in respect to the background.

In SCENIC workflow that we used to predict regulon (matching a TF to co-expressed regulated genes) activity, the input genes are first filtered by their availability in RcisTarget database, and then individual regulons are constructed from scRNA-seq data with GENIE3. In other words, the regulons are refined via RcisTarget by keeping genes that contain the respective TF binding motifs from the current analyzed dataset. Subsequently, statistical method AUCell is applied to score individual cells by assessing for each TF separately to define if the co-expressed genes are enriched in the top quantile of the cell signature. Since the AUCell scoring method is ranking-based, AUCell is independent of the gene expression units and the normalization procedure.

It should be noted that, while DoRothEA and D-AUCell rely on independent regulons, the SCENIC networks are constructed from the same dataset they are applied to. This poses the risk of overfitting. Please consider other tools for confirmation.

We thank the reviewer for the suggestion, and we have addressed this as follows:

Since SCENIC networks are constructed from the same scRNA-seq dataset they applied to, this poses the risk of overfitting. Thus, we have performed DoRothEA^{1, 2}, which relies on independent TF-targeted gene interactions (regulon activity) curated from various resources, such as literatures, ChIP-seq peaks, TF binding motifs and gene expression inferred interactions, to compare its results with SCENIC. Based on the number of supporting evidences, an interaction confidence level is calculated, ranging from A to E, with A being the most confidence interactions and E the least. VIPER is a statistical method that used in combination with DoRothEA to estimate TF activities from scRNA-seq expression data.

This approach emphasizes the potential importance of some of the TFs seen previously in our SCENIC analysis, prominently, HIF1A, STAT1 and FOSL2, as well as SMAD3. We have added this analysis as Figure S17, and as mentioned below also on the basis of this and the other additional rigor added in response to the comments below expanded the discussion on these TFs.

1. Garcia-Alonso L, Holland CH, Ibrahim MM, Turei D, Saez-Rodriguez J. Benchmark and integration of resources for the estimation of human transcription factor activities. *Genome Res* **29**, 1363-1375 (2019).
2. Holland CH, et al. Robustness and applicability of transcription factor and pathway analysis tools on single-cell RNA-seq data. *Genome Biol* **21**, 36 (2020).

2. All genes showing expression in at least one cell seems very permissive as SCENIC and other regulatory analysis tools are easily influenced by the gene input. Please try with more robust cutoff and show significance of this analysis. Furthermore, the number of cells in each sub cluster are dramatically different. Unequal variance of cell number will influence the statistics to derive meaningful motifs. Please try using similar cell numbers (e.g. downsampling) and see if the conclusions are the same.

Response: To set a more robust cutoff, we adjusted the filtering to keep genes 1) with at least 6 UMI counts across all cells, and 2) detected in at least 1% of cells. We added figure S14 with SCENIC analyzing these filtered genes matched in RiciTarget database.

Furthermore, the number of cells in each sub cluster are dramatically different. Unequal variance of cell number will influence the statistics to derive meaningful motifs. Please try using similar cell numbers (e.g. downsampling) and see if the conclusions are the same.

Response: We agree that the number of cells in subclusters 3_WIF1+ and 4+PRSS23+ (673 and 743 cells, respectively) are quite different from the number of cells in subcluster 4_SFRP4+ (73 cells). However, removing the majority of cells from 3_WIF1+ and 4+PRSS23+ subclusters to compensate for the smaller number of cells in SFRP4+ would likely result in excluding representative information, lowering the power of the AUCell analysis for ranking the TF regulons. Thus, anticipating there would be some variability when randomly downsampling 73 cells from 3_WIF1+ and 4+PRSS23+ subclusters, we compared multiple runs by SCENIC. Figure S15: Three independent sampling (73 cells for all three clusters) were performed followed with the same filtering set above for running SCENIC workflow. Regulons up in SFRP4+ showed before, such as FOXP1, IRF7 and RUNX2, were showed up in 2 out of 3 downsampling, while other regulons like STAT1 that showed up twice here wasn't observed before without downsampling. We have added these results as Figure S15.

Finally, we would like to emphasize the results in Figure 8E, showing that several TFs predicted to be important through the regulon analyses, are upregulated at the gene expression level. While we realize increased gene expression does not necessarily mean more protein or enhanced TF activity, the combined data showing enhanced predicted motif activation along with together are supportive of the importance of these highlighted TFs in regulating myofibroblast differentiation.

3. Why does SMAD3 only come up only when restricted genes are used as input? Again, this may be easily influenced by the input number and how the background is defined. To ensure SMAD3 is not an artifact,

functional validation in culture fibroblasts or ChIP-seq validation (e.g. Cut & Tag) from tissue samples would substantially prove this.

This question and comment appear less pertinent in view of the reanalysis using DoRothEA suggested in comment 2, reinforcing the importance of SMAD3 in regulating the transcriptome of SSc fibroblasts. This was seen consistently across the different levels of DoRothEA confidence, including the highest level (level A). The targets of Smad3 using DoRothEA included well-known Smad3 targets such as COL1A1, SERPINE1 and TNC. Although we agree that Cut & Tag is an exciting approach to defining TF binding with low input cell number, we have not yet been able to use this successfully in single fibroblast from digested skin tissues. Instead, in order to provide greater confidence in the predicted Smad3 targets from SCENIC, we have identified a TGFβ regulated gene by microarray analysis of TGFβ treated fibroblasts that is included in the otherwise less readily recognizable SMAD3 targets found by SCENIC, in particular CHAC1. We have added a Figure showing that CHAC1 expression induced by TGFβ in dermal fibroblasts from control subjects and SSc patients is inhibited by pre-treating the cells with SIS3, an inhibitor of Smad3 phosphorylation. This result along with the results using DoRothEA together strongly validate the results of SCENIC.

In light of the strengthening of the section of the manuscript predicting TFs regulating the SSc fibroblast transcriptome, based on our responses to these helpful reviewer's comments, we have added briefly to the discussion, highlighting literature implicating some of the TFs showing up consistently in these models in SSc through genetic analyses, murine models of skin fibrosis and/or in vitro studies.

2. The immune staining is still not convincing. The antibodies used in past studies differ from the one that is presented in this manuscript. Therefore a proper control is necessary.

In this paper:

polyclonal rabbit SFRP4 (1:500;153287-1-AP; Proteintech, USA)

In J Invest Dermatol. 2008;128(4):871-81,

rabbit polyclonal affinity-purified anti-SFRP4 antibody (1:70 dilution; kindly provided by Dr R. Friis)

In J Invest Dermatol. 2018;138(4):802-10

No SFRP4 was tested.

Response: We have added a supplemental figure (Figure S10C) showing the SFRP4 staining in the deep reticular dermis in SSc skin and not in normal skin, using the same antibody used in Figure 4B.

Unfortunately, the antibody previously provided by Dr. R. Friis is no longer available (he retired several years ago and the antibody has been lost).

3. There is a big discrepancy between the single cell RNA-seq data where SFRP4 is a small percentage of SFRP2, the immune-staining seem to suggest nearly 100% overlap between the two markers. This either indicates problems with the staining background or single cell data is not representative of cell population. Please clarify.

Response: This is clarified in the supplemental figure, mentioned in response to comment 2 (Figure S10C). In Figure 4B we show the overlapping staining between SFRP2, SFRP4 and SMA (smooth muscle actin). The co-staining with these three markers is a key observation in the manuscript, since the other points (described following) have been made in other publications.

To amplify this point further, as it is a key finding described in the manuscript, myofibroblasts are highly variable in number between different SSc patients, their number correlating with the intensity of skin fibrosis as assessed by the modified Rodnan skin score (Arthritis Rheum. 2006;54(11):3655-60; PMID: 17075814). I have personally examined hundreds of SMA stained SSc skin biopsies, SMA staining highly elongated cells in the deep reticular dermis, in mildly affected skin primarily staining cells at the interface with the fascia and in more severe cases staining cells progressively higher in the dermis. I have never seen myofibroblasts in healthy control skin. SFRP2 staining fibroblasts, on the other hand, make up the majority

population of fibroblasts in normal skin (J Invest Dermatol. 2018;138(4):802-10. PMID: 29080679). Like myofibroblasts, reticular SFRP4-staining fibroblasts are seen only in SSc skin (J Invest Dermatol. 2008;128(4):871-81. PMID: 17943183). Like myofibroblasts, SFRP4 staining reticular fibroblasts are highly variable, their number correlating with the modified Rodnan skin score. In retrospect I should have realized that SMA and SFRP4 were staining the same cells. However, this only became clear with the scRNA-seq data and subsequently we were able to confirm not only that SFRP4 and SMA cells are the same, but also that SFRP4 (and SMA) staining cells also stain with SFRP2, strongly pointing to their cellular origin. However, most SFRP2 staining cells do not stain with SMA or SFRP4, and SFRP2 fibroblasts can easily be seen to stain throughout the dermis in normal skin.

4. Figure 8 panel E and F should be swapped.

Response: Thanks, we have corrected the figure legend for this figure.

REVIEWER COMMENTS

Reviewer #2 (Remarks to the Author):

The authors have addressed the comments raised by this reviewer. However, many of the analyses seem arbitrary. For example, what is the justification for downsampling cells only for three times? Bootstrapping (or permutation tests) to derive significance may be more suited in this case. Further, filtering cells with 6 UMI counts also seems arbitrary. How did the authors derived this number? Please consider taking a range of UMI values or percentages (1-10 UMI or 1 to 5%, for example) and quantify the occurrence to which the target motif is observed. Finally, SMAD3 is known to have multiple targets but why was only CHAC1 selected? How did the authors derive this target and what about the others?

These comments do not question the conclusion made in this report however the statistical methods to showcase these reports, especially when analyzing large-scale single cell genomics data, should be devoid of authors' subjective and arbitrary threshold that fits the preconceived notion of expected biology.

POINT-BY-POINT RESPONSE TO REVIEW:

“The authors have addressed the comments raised by this reviewer. However, many of the analyses seem arbitrary. For example, what is the justification for downsampling cells only for three times? Bootstrapping (or permutation tests) to derive significance may be more suited in this case. Further, filtering cells with 6 UMI counts also seems arbitrary. How did the authors derived this number? Please consider taking a range of UMI values or percentages (1-10 UMI or 1 to 5%, for example) and quantify the occurrence to which the target motif is observed. Finally, SMAD3 is known to have multiple targets but why was only CHAC1 selected? How did the authors derive this target and what about the others?”

These comments do not question the conclusion made in this report however the statistical methods to showcase these reports, especially when analyzing large-scale single cell genomics data, should be devoid of authors’ subjective and arbitrary threshold that fits the preconceived notion of expected biology.”

We appreciate the reviewer’s feedback, but we believe that most of these latest comments would have more merit if we were developing a new computational method to analyze scRNA-seq data. If that was the case, then we should had performed an exhaustive search of all parameters used, including number of UMIs, number of downsamplings, etc. By contrast, we’d like to mention that the value of our paper is in the biological knowledge it generates. Specifically, in that it shows the origin of myofibroblasts in SSc skin disease, it identifies definitive gene expression and immunohistochemical markers, it shows that these cells differentiate from a fibroblast subpopulation found in normal human dermis and it indicates by both increased level of expression and SCENIC that certain transcription factors are particularly important in the process of myofibroblast differentiation. In this context, it is of little importance how we derived the computational predictions, as long as the results are sound and supported by the wet lab experiments. Or, to put it in the reviewer’s own words: “*These comments do not question the conclusion made in this report*”.

With this framework in mind, we provide a point-by-point response to reviewer’s comments:

1) What is the justification for downsampling cells only for three times? Bootstrapping (or permutation tests) to derive significance may be more suited in this case.

In order to test if the difference of cell number among clusters of 3 (WIF1+), 4 (PRSS23+), and 4 (SFRP4+ subset; 673, 743, 73 cells, respectively) would affect the SCENIC analysis in Figures 4F, S14 and S15, we downsampled, using R function ‘sample’ to randomly select 73 cells from cluster 3 (WIF1+) and 4 (PRSS23+) respectively. We have now downsampled and performed SCENIC analysis 40 times with the different pools of cells. In the new Fig. 8F, we show the most consistent (38 non-redundant out of 40 runs) regulons for each population across the SCENIC iterations. The activities of IRF7, STAT1 and CREB3L1 were increased consistently in the cluster 4, SFRP4 subpopulation of cells. The original Figure 8F has been moved to a supplemental file (Figure S14) and the .

2) Filtering cells with 6 UMI counts also seems arbitrary. How did the authors derived this number?

Filtering the cells to 6 UMI counts was also something we added at the request of this reviewer. In our initial analysis we had not excluded any transcript (essentially, UMI=1 was our cutoff), but to comply with the reviewer’s request, we applied the default UMI cutoff value of SCENIC, the popular software we used. We also like to point out that UMI=1 was proven to be sufficient according to a recent study in *Nat Commun* (2). Regardless, we performed additional analysis to examine the effect of UMI threshold to the reduction of transcript number. In the figure below, we see that beyond UMI=5 there is not much reduction in transcript number. For those reasons (UMI=6 is the SCENIC default cutoff, similar studies using UMI=1, no reduction in transcripts after UMI=5 cutoff) we believe that the two analyses we performed provide sufficient justification for the choice of UMI cutoffs.

3) Please consider taking a range of UMI values or percentages (1-10 UMI or 1 to 5%, for example) and quantify the occurrence to which the target motif is observed.

Please see our previous response and my opening comments. Using different thresholding for UMIs or percent of cells expressing the genes has no scientific utility, and was not the approach currently used (e.g., in the recent *Nature* publication). In any case, we'd like to point out that the reviewer agrees that the additional analyses she/he proposes “do not question the conclusion made in this report.”

4) Finally, SMAD3 is known to have multiple targets but why was only CHAC1 selected? How did the authors derive this target and what about the others? ...especially when analyzing large-scale single cell genomics data, should be devoid of authors' subjective and arbitrary threshold that fits the preconceived notion of expected biology

The analyses by DoRothEA (3), which was asked for by this reviewer, actually served to reinforce a possible role for SMAD3, also supported by enormous literature, so we don't understand why the reviewer suggests that we are arbitrarily focusing on this TF. SCENIC in its first step analyzes gene regulatory networks, i.e., it looks at transcription factors regulated in each cell and then looks at all of the genes regulated in a similar fashion across the set of cells. This is an extraordinarily powerful process, but its performance depends on the (increased) expression level of the transcription factor (in certain cells). It is inherently easier to see co-regulated, putative target genes when a transcription factor is expressed in many cells (or at different levels), than when it is expressed at low levels. SMAD3 is expressed in relatively few cells in our dataset. SCENIC reveals unexpected associations between the TFs and their predicted targets, but in our experience these genes are not commonly recognized in the literature. This appears likely because they are expressed at very low levels. Despite this limitation, SCENIC identified SMAD3 and several target genes. However, these target genes were also expressed at very low levels (in very few cells). Therefore, we focused on only one of these genes, CHAC1, because: 1) we were able to show that it was regulated by TGF β as well as downregulated by Smad3 inhibition and 2) because it was detectable by RT-PCR in our cells. This allowed us to define within the SCENIC gene regulatory network a direct target of SMAD3 – CHAC1, which we show in response to the previous review as regulated directly by SMAD3 phosphorylation. To reiterate, CHAC1 was chosen because other genes that were predicted as SMAD3 targets by SCENIC either failed to be regulated by TGF- β or as in most cases were expressed at too low of levels to be reliably detected. We would emphasize that this does not mean they are not important targets, but rather that their expression levels are quite low. DoRothEA on the other hand, looks at genes known from past literature and datasets to be downstream of the TF. These genes are relatively easily detected, and thus the DoRothEA curated database has been used by others to verify SCENIC results.

Recently Garcia-Alonso et al (3) reviewed the different approaches to identifying “the set of target genes directly regulated by a TF:” 1) manually curated from literature; 2) direct TF-DNA binding assays; 3) computational predictions of TF–target interactions, i.e., position weight matrices (PWMs); and 4) gene regulatory networks that assume TF expression levels regulate targeted genes. SCENIC uses approach 4) in its initial screening giving considerable weight to this approach. The resulting gene regulatory network is then filtered on CisTarget, a database incorporating results used in methods 1-3. Each of these approaches have merit, but argue in favor of using complementary methods to support SCENIC results.

To this end, we here respond to this critique using the same approach used in a recent *Nature* paper (1), in which RCisTarget (R implementation of CisTarget, the same database used by SCENIC) was used to identify putative important TF, but then DoRothEA regulon genes were used to build an AP-1 module for probing the role of the TF in the dataset. As this manuscript is on a similar topic in renal disease, was submitted after ours, and currently published, it is concurrent and could therefore serve as a reasonable benchmark for the use of these bioinformatics methodologies. I have reproduced the methods section of this manuscript following, underlining the most relevant part to our further analysis of our SCENIC results.

“To obtain transcription factor scores in distal and proximal regions, we used the top 200 marker genes for fibroblast, pericyte and myofibroblast cell clusters as input gene lists to RCisTarget⁷⁷. We followed the RCisTarget Vignette to perform the analysis with default parameters (available at <https://bioconductor.org/packages/release/bioc/vignettes/RcisTarget/inst/doc/RcisTarget.html>). To quantify AP-1 expression, we used all Jun and Fos genes as a geneset and applied the same method to obtain an AP-1 score as we did for ECM score. To quantify AP-1 activity (defined as the expression of putative target genes^{78,79}, we defined AP-1 target genes according to the Dorothea regulon database^{63,80} and applied the same method as ECM score to obtain a single cell AP-1 activity score”

Thus, as an alternative approach to further assess the SMAD3 targets, and the same approach used by the recently published *Nature* paper on myofibroblasts in renal fibrosis (1), we have now examined expression of SMAD3 target genes in our single cell data, using the DoRothEA regulon database. Namely, we created a SMAD3 activity score using the 33, DoRothEA level A, SMAD3 target genes and applied this to our fibroblast data using the Seurat AddModuleScore, we plotted these activity scores on scRNA-seq UMAP feature plots. This plot supports a role for Smad3 in regulation of the transcriptomes of the SSc SFRP2+ cells as they mature into myofibroblasts. We have added this observation as supplemental Figure S19.

In addition, we have carried out RNA-seq experiments to test genes in myofibroblasts inhibited by SMAD3 siRNA compared to non-targeting siRNA and HRPT1 (control) treated siRNA (Table S10), and selected genes based on two criteria to create SMAD3 activity scores. Namely, we derived SMAD3 activity scores from: 1) SMAD3 siRNA treated myofibroblasts, filtered for absolute gene expression of non-targeting control siRNA >50 TPM, showing expression of SMAD3 siRNA treated cells of less than 0.8 of non-targeting control siRNA, and excluding genes showing expression less than 0.7 in HRPT1 (control) siRNA treated cells compared to non-targeting control siRNA (415 genes); or 2) SMAD3 siRNA treated myofibroblasts, filtered for absolute gene expression in non-targeting control treated siRNA cells of >100 TPM, showing expression of SMAD3 siRNA treated cells of less than 0.7 of non-targeting control siRNA, and excluding genes showing expression less than 0.7 in HRPT1 (control) siRNA treated cells compared to non-targeting control siRNA (74 genes). Using the Seurat AddModuleScore, we plotted these activity scores on scRNA-seq UMAP feature plots, adding these as Figure S20. Both of these data-driven SMAD3 activity scores indicate increased SMAD3 activity in SSc SFRP2+ fibroblasts and SScSFRP2+SFRP4+ myofibroblasts even more clearly than the activity score derived from DoRothEA.

1. C. Kuppe *et al.*, Decoding myofibroblast origins in human kidney fibrosis. *Nature* **589**, 281-286 (2021).
2. M. J. Zhang, V. Ntranos, D. Tse, Determining sequencing depth in a single-cell RNA-seq experiment. *Nat Commun* **11**, 774 (2020).
3. L. Garcia-Alonso, C. H. Holland, M. M. Ibrahim, D. Turei, J. Saez-Rodriguez, Benchmark and integration of resources for the estimation of human transcription factor activities. *Genome Res* **29**, 1363-1375 (2019).

REVIEWERS' COMMENTS

Reviewer #2 (Remarks to the Author):

The reviewer appreciates thorough explanation and justifications of the methods used in the revised version of the manuscript and has no further comments.